# Therapeutic Potential and Mechanisms of Novel Simple O-Substituted Isoflavones against Cerebral Ischemia Reperfusion

**DOI:** 10.3390/ijms231810394

**Published:** 2022-09-08

**Authors:** Shu-Er Yang, Jin-Cherng Lien, Chia-Wen Tsai, Chi-Rei Wu

**Affiliations:** 1Department of Beauty Science and Graduate Institute of Beauty Science Technology, Chienkuo Technology University, Changhua City 500020, Taiwan; 2School of Pharmacy, China Medical University, Taichung 404328, Taiwan; 3Department of Nutrition, China Medical University, Taichung 404328, Taiwan; 4Department of Chinese Pharmaceutical Sciences and Chinese Medicine Resources, China Medical University, Taichung 404328, Taiwan

**Keywords:** isoflavones, biosynthesis, metabolism, cerebral ischemia reperfusion

## Abstract

Isoflavones have been widely studied and have attracted extensive attention in fields ranging from chemotaxonomy and plant physiology to human nutrition and medicine. Isoflavones are often divided into three subgroups: simple O-substituted derivatives, prenylated derivatives, and glycosides. Simple O-substituted isoflavones and their glycosides, such as daidzein (daidzin), genistein (genistin), glycitein (glycitin), biochanin A (astroside), and formononetin (ononin), are the most common ingredients in legumes and are considered as phytoestrogens for daily dietary hormone replacement therapy due to their structural similarity to 17-β-estradiol. On the basis of the known estrogen-like potency, these above isoflavones possess multiple pharmacological activities such as antioxidant, anti-inflammatory, anticancer, anti-angiogenetic, hepatoprotective, antidiabetic, antilipidemic, anti-osteoporotic, and neuroprotective activities. However, there are very few review studies on the protective effects of these novel isoflavones and their related compounds in cerebral ischemia reperfusion. This review primarily focuses on the biosynthesis, metabolism, and neuroprotective mechanism of these aforementioned novel isoflavones in cerebral ischemia reperfusion. From these published works in in vitro and in vivo studies, simple O-substituted isoflavones could serve as promising therapeutic compounds for the prevention and treatment of cerebral ischemia reperfusion via their estrogenic receptor properties and neuron-modulatory, antioxidant, anti-inflammatory, and anti-apoptotic effects. The detailed mechanism of the protective effects of simple O-substituted isoflavones against cerebral ischemia reperfusion might be related to the PI3K/AKT/ERK/mTOR or GSK-3β pathway, eNOS/Keap1/Nrf-2/HO-1 pathway, TLRs/TIRAP/MyD88/NFκ-B pathway, and Bcl-2-regulated anti-apoptotic pathway. However, clinical trials are needed to verify their potential on cerebral ischemia reperfusion because past studies were conducted with rodents and prophylactic administration.

## 1. Introduction

Stroke is the second leading cause of death globally, affecting approximately 13.7 million people each year, of which 6 million die. According to the definition of the World Health Organization, it involves rapidly developing clinical signs of focal (or global) disturbance of cerebral function, with symptoms lasting 24 h or longer, or leading to death, with no apparent cause other than being of vascular origin [1,2]. On the basis of its causes, stroke can be categorized into ischemic stroke and hemorrhagic stroke. Ischemic stroke is the common form accounting for between 80 and 85% of all strokes, while the other form, hemorrhagic stroke, accounts for the remaining 15–20%, of which subarachnoid hemorrhage accounts for 3% of all stroke types [3,4]. In general, most strokes occur after age 50. When cerebrovascular events occur between 20 weeks of fetal life and 28 days of postnatal life, they are called perinatal strokes. Approximately between 80 and 90% of perinatal strokes acutely occur in the neonatal period, termed neonatal arterial ischemic strokes [5]. Ischemic stroke is mainly caused by blood clots that have formed within the cerebral artery, which drop cerebral blood flow below 10 mL/100 g/min and consequently block the blood supply in certain areas of the brain for a few minutes. Hemorrhagic stroke is mainly caused by the rupture of blood vessels, causing bleeding in the brain, thus, inhibiting oxygen and nutrients from reaching brain cells. The acute clinical symptoms of ischemic stroke include trouble walking or problems with balance; trouble seeing in one or both eyes; numbness or weakness in the face, arm, or leg; trouble speaking or understanding speech; severe headache with no known cause. Because the etiology is thrombosis, initial treatment drugs (3-h time window of treatment) include thrombolytic drugs such as rtPA, streptokinase, urokinase, alteplase, and tenecteplase. Some adjuvant drugs, such as aspirin, nitric oxide synthase (NOS) inhibitors, and antioxidants, are also used. After the 3-h time window of treatment (from days to weeks or from months to years after stroke), maintenance drugs involve anticoagulants, antiplatelets, statins, and neuroprotectants [6].

This interruption from ischemic stroke leads to brain infarction, hypoxia, and hypoglycemia of cerebral nerve cells. Consequentially, the ischemic cascade is further triggered. Due to hypoxia and hypoglycemia, brain energy metabolism is altered, including a lack of ATP synthesis and an accumulation of lactate. Subsequently, ion pumps (Na^+^/K^+^-ATPase and Ca^2+^/H-ATPase) malfunction occurs, resulting in membrane potential depolarization and calcium ion (Ca^2+^) overload [7]. Membrane depolarization leads to the excess release of excitotoxic amino acids (especially glutamate), which trigger excitotoxicity. This excitotoxicity induces a cascade of neurotoxicity that ultimately leads to necrotic and apoptotic cell death, as well as the loss of neuronal function. This process sequentially activates postsynaptic and extra-synaptic glutamate receptors, calcium ion influx, oxidative stress, mitochondrial dysfunction, endoplasmic reticulum stress, inflammation, and the activation of intracellular calcium-dependent enzymes and proteins (including phospholipase A2 and C (PLA2 and PLC), cyclooxygenase, nitric oxide synthase, protein kinase C (PKC), proteases and endonucleases) [8,9,10,11,12,13]. Oxidative stress is one of the important parts of the excitotoxic cascade. Overactivation of glutamate receptors causes the activation of the pro-oxidant enzymes xanthine oxidase and nicotinamide adenine dinucleotide phosphate oxidase (NADPH oxidase) to induce the excess production of reactive oxygen species (ROS). Combined with reactive nitrogen species (RNS) generated by NOS activation, excess ROS and RNS cause enzyme inactivation, lipidic peroxidation, and consequent membrane damage and DNA alterations [14,15,16]. Neuroinflammation is another important part of the excitotoxic cascade. Excess glutamate and ROS activate resident microglia and astrocytes, together with infiltrated T lymphocytes, neutrophils, and macrophages. The activation of these immune cells promotes the release of multiple inflammatory factors, such as cytokines, chemokines, and free radicals [17]. These above processes occur in series and in cycles. Lastly, cerebral nerve cell death in this ischemic area and the decrease in cerebral nerve functions, such as motor and neurobehavior function, depending on the physical function of the ischemic area, also occur. On the other hand, they also disrupt the integrity of the blood–brain barrier (BBB), thus, enlarging the infarction volume and aggravating the damage of brain nerve cells and intracellular signaling dysregulation [18,19,20]. For an indefinite time after cerebral ischemia, reperfusion might occur to increase blood supply to the ischemic area. In addition, thrombolytic therapy can promote the production of reperfusion [7,21]. Furthermore, it can promote oxidative stress and inflammation, causing secondary damage to brain tissues. Hence, the brain damage caused by stroke is often called cerebral ischemia reperfusion (IR) injury, which include two stages: primary cerebral ischemic injury and secondary cerebral reperfusion injury. Cerebral IR injury is a vicious cycle, and its pathophysiological mechanisms include the failure of brain energy metabolism, disruption of ion homeostasis, excitotoxicity, calcium overload, oxidative stress, endoplasmic reticulum stress, mitochondrial dysfunction, and inflammation. Then, apoptotic- and autophagy-related enzymes and genes are activated to cause neuronal necrosis and apoptosis, subsequently decreasing intracellular signaling transduction and cerebral nerve functions, such as motor and neurobehavior function [8,9,13,14,15,17,22,23]. Due to the causes and pathophysiological cascades of IR, oxygen deprivation (hypoxia), oxygen and glucose deprivation (OGD), and treatment with excitotoxic toxins (such as glutamate or N-methyl-D-aspartate (NMDA)) are commonly used in in vitro research models. They can induce the pathophysiological phenomena of IR such as energy failure, excitotoxicity, calcium overflow, oxidative stress, endoplasmic reticulum stress, mitochondrial dysfunction, and neuronal apoptosis. In vivo research models, global cerebral ischemia reperfusion, four-vessel occlusion (4-VO), and middle cerebral artery occlusion (MCAO) are often used in gerbils or rodents [24,25,26,27]. Many natural products have been evaluated for their protective effects on IR using in vitro and in vivo experimental IR models based on the pathological mechanism of IR, and isoflavones are one of them.

Plant secondary metabolites (natural products) are produced by the plant cells in plant–environment interactions through some metabolic pathways derived from the primary metabolic pathways. As the products of plant–environment interactions, some plant secondary metabolites function as phytoalexins or phytoantibiotics to protect plants from pathogens. Later on, researchers found that they had antibacterial, antifungal, and antiviral activities in humans. Scientists have continuously pointed out that plant secondary metabolites play important roles in plant physiology, plant pathology, human nutrition, and medicine. Plant secondary metabolites are represented by a variety of low molecular weight organic compounds. According to their structure, plant secondary metabolites are classified into several classes, such as terpenoids, glycosides, phenolics, and alkaloids [28,29]. Flavonoids are the largest and most studied subclass of the phenolics class among many plant secondary metabolites. According to the characteristics of chemical structure, flavonoids can be divided into two main groups: the flavonoids group (mainly including flavanones, flavanonols, flavones, flavonols, flavan-3-ols, flavan-4-ols, and anthocyanidins) and the isoflavonoids group (mainly including isoflavones, isoflavans, pterocarpans, and rotenoids). The structure of flavonoids comprises the 2-phenylchroman skeleton, which is a linear C6-C3-C6 skeleton derived from a phenylpropanoid (C6-C3) starter and three C2 elongation units. However, the structure of the isoflavonoids group comprises the 3-phenylchroman skeleton, which is derived from the 2-phenylchroman skeleton via an aryl migration reaction (Figure 1) [30]. Isoflavonoids have been widely studied and have attracted substantial attention in fields ranging from chemotaxonomy and plant physiology to human nutrition and medicine. In chemotaxonomy, isoflavonoids are recognized as chemosystematic markers due to their presence and distribution in different plants. In general, isoflavonoids are mostly found in legumes (Fabaceae family), such as *Astragalus* spp., *Dalbergia odorifera* T. C. Chen, *Glycyrrhiza* spp. (Licorice), *Glycine max* (L.) Merr. (soybean), *Hedysarum multijugum* Maxim, *Iris* spp., *Lupinus* spp., *Medicago* spp. (barrel medic), *Pueraria montana* var. *lobata* (Lour.) Merr., *Sophora flavescens* Aiton, and *Trifolium* spp. (clover). Occasionally, isoflavonoids can be found in 31 non-leguminous families, such as *Asteraceae, Chenopodiaceae, Iridaceae, Moraceae, Myristicaceae,* and *Nyctaginaceae* [31,32,33,34,35]. In plant physiology, isoflavonoids are considered as phytoalexins or phytoantiobiotics due to their antifungal and insecticidal properties. In human nutrition and medicine, isoflavonoids are known as phytoestrogens due to their structural similarity with 17-β-estradiol. So far, approximately 1600 isoflavonoids have been identified, seven subclasses (isoflavones, isoflavans, isoflavanones, isoflavanols, pterocarpans, coumestans, and rotenoids) are mainly mentioned, which are generated through the biosynthetic modification of core (general or central) phenylpropanoid pathways and downstream flavonoid/isoflavonoid branch pathways. In general, isoflavones are the most important and largest subgroup of isoflavonoids, and they often serve as precursors of other isoflavonoid phytoalexins such as isoflavans, pterocarpans, rotenoids, and coumestans due to their core isoflavonoid scaffold. According to the difference in the substitutions and the position of substitutions on the basic structural skeleton, isoflavones are often divided into three subgroups: simple O-substituted derivatives (i.e., hydroxy, methoxy, methylenedioxy, etc.), prenylated derivatives (i.e., prenyl, isoprenyl, etc.), and glycosides [31,32,33,34,35]. Among these three subgroups of isoflavones, simple O-substituted isoflavones such as daidzein, genistein, glycitein, biochanin A, and formononetin are the most common ingredients in the legumes used as daily dietary supplements. The glycosides of simple O-substituted isoflavones (daidzin, genistin, glycitin, ononin, and astroside) are more abundant than these aglycones in the legumes. Due to their structural similarity with 17-β-estradiol, these above isoflavones (simple O-substituted isoflavones, as well as their glycosides and metabolites) are considered to possess estrogen-like properties via binding to estrogen receptors (ERs), leading to the activation of estrogen-related receptor response elements (ERRE) located on the inner side of the nuclear membrane, and, in turn. promoting gene transcription processes. However, simple O-substituted isoflavones usually possess a higher affinity for ERs than their glycosides. As for the affinity of their metabolites for ERs, there is a large degree of variability. For example, the affinity of equol for ERs is greater than that of its parent isoflavone-daidzein, and almost equivalent to that of genistein. However, O-desmethylangolensin (O-DMA) has a weaker affinity for ERs almost equivalent to its parent isoflavone-daidzein. Hence, these simple O-substituted isoflavones and some metabolites are recognized as phytoestrogens for use as dietary estrogen supplements in alternative hormone replacement therapy [36,37,38,39]. Many researchers have studied the various pharmacological activities of these novel simple O-substituted isoflavones (daidzein, genistein, glycitein, formononetin, and biochanin A), their glycosides (including daidzin, genistin, glycitin, ononin, and astroside), and their metabolites (equol and O-DMA). These novel simple O-substituted isoflavones, as well as their glycosides and metabolites, possess many pharmacological activities similar to estrogen, including antioxidant, anti-inflammatory, anticancer, anti-angiogenetic, hepatoprotective, antidiabetic, antilipidemic, anti-osteoporotic, and neuroprotective activities. Some review studies on versatile pharmacological potential, anticancer, antidiabetic, and antilipidemic activities have been presented for individual isoflavones, such as soy isoflavones, genistein, genistin, formononetin, biochanin A, equol, and O-DMA [36,39,40,41,42,43,44,45,46,47,48,49,50,51,52,53,54,55,56,57,58]. However, there are very few review studies on the protective effects of these novel O-substituted isoflavones in cerebral IR. This review primarily focuses on the biosynthesis, metabolism, and protective mechanism of the aforementioned novel O-substituted isoflavones (including their glycosides and metabolites) (Figure 2) in cerebral IR.

## 2. The Biosynthesis Pathway of Isoflavones

The biosynthesis pathway and related enzymes of isoflavones have been investigated extensively using biochemical and genetic methods. Nearly all enzymes in isoflavone biosynthesis have been isolated, and their structure and functional characterization are understood. Isoflavonoids are synthesized through the core phenylpropanoid pathway and downstream flavonoids/isoflavonoids biosynthetic branch pathway (Figure 3) [59,60].

The core phenylpropanoid pathway is an important metabolic pathway in plant–environment interactions, which produces secondary metabolites, such as flavonoids and isoflavonoids, from the main substrates, such as phenylalanine or tyrosine. In the initial step of the core phenylpropanoid pathway, phenylalanine (produced from the shikimate pathway) is deaminated by phenylalanine ammonia-lyase (PAL) into cinnamic acid, a precursor of various phenylpropanoids. Then, cinnamic acid is hydroxylated into *p*-coumarate by cinnamate 4-hydroxylase (C4H). In the final step of the core phenylpropanoid pathway, *p*-coumarate was converted to *p*-coumaroyl-CoA by 4-coumarate CoA-ligase (4CL). *p*-Coumaroyl CoA is at the junction of the downstream biosynthetic branch pathways leading to flavonoid/isoflavonoid/pterocarpan biosynthesis, to flavonoid/anthocyanin/condensed tannin biosynthesis, or to lignin/lignan biosynthesis [61,62].

Chalcone synthase (CHS) is the first committed enzyme of the flavonoid/anthocyanin/condensed tannin and flavonoid/isoflavonoid branch pathway. CHS catalyzes the formation of a tetrahydrochalcone–naringenin chalcone (4,2′,4′,6′-tetrahydroxychalcone) from one molecule of *p*-coumaroyl-CoA (produced from the core phenylpropanoid pathway) with three malonyl-CoA molecules (produced from acetate pathway) through the stepwise condensation. This naringenin chalcone is catalyzed by chalcone isomerase (CHI) to form one flavanone–naringenin (5,7,4′-trihydroxy-flavanone) through intramolecular cyclization. This step is recognized as one branch of the flavonoid/isoflavonoid/pterocarpan biosynthesis pathway, but it is a common step in most plants. The other branch of the flavonoid/isoflavonoid/pterocarpan biosynthesis pathway is carried out via the co-action of CHS and legume-specific chalcone reductase (CHR) in most legume species. One trihydroxychalcone isoliquiritigenin (4,2′,4′-trihydroxychalcone) is formed from one molecule of *p*-coumaroyl-CoA and three malonyl-CoA molecules, which are catalyzed by CHS and CHR. Then isoliquiritigenin is also transformed into one flavanone liquiritigenin (4′,7-dihydroxyflavanone) by CHI. The two flavanone intermediates-naringenin and liquiritigenin are successively converted into 2-hydroxyisoflavanones (2-hydroxy-2,3 dihydrogenistein and 2,7,4′-trihydroxyisoflavanone) by isoflavone synthase (IFS, also called 2-hydroxyisoflavanone synthase (2-HIS)) through the aryl migration reaction of the aromatic B-ring from the C-2 to the C-3 position and hydroxylation at the C-2 position. Then the intermediate 2-hydroxyisoflavanones are dehydrated by 2-hydroxyisoflavanone dehydratase (HID) to form the corresponding isoflavones (genistein and daidzein) through the formation of a double bond between the C-2 position and C-3 position of the C-ring. On the other hand, liquiritigenin is first converted into 6-hydroxyflavanone (6,7,4′-trihydroxy flavanone) by flavonoid 6-hydroxylase (F6H). Then, 6,7,4′-trihydroxyflavanone is gradually transformed into 6-hydroxydaidzein by IFS and HID [59,60].

These two isoflavones (genistein and daidzein) are precursors of other isoflavone derivatives because they have a core isoflavone scaffold. They undergo a variety of modification reactions, such as methylation, acetylation, hydroxylation, glycosylation, and malonylation. Among these modification reactions, O-methylation often occurs in the early stage of the flavonoid/isoflavonoid/pterocarpan biosynthesis pathway. This modification increases the lipophilicity of isoflavones and reduces their chemical reactivity, leading to an increase in their antimicrobial activities and their use as phytoalexins for disease resistance in the plants. The most common site of methylation is at the 4′ position of isoflavones. Hydroxyisoflavanone 4′-specific O-methyltransferase (HI4′OMT) catalyzes the transfer of a methyl group to a specific hydroxyl group of these two isoflavones (genistein and daidzein), forming biochanin A and formononetin. Biochanin A and formononetin are also produced from the catalysis of 2-hydroxy isoflavanone (2-hydroxy-2,3 dihydrogenistein and 2,7,4′-trihydroxyisoflavanone) by HI4′OMT and HIS. On the other hand, 6-hydroxydaidzein is also modified by hydroxyisoflavanone O-methyltransferase (HIOMT) to form glycitein. Additionally, glycosylation is the most frequently occurring and most extensive modification of isoflavones. This modification often reflects the interaction between plants and symbiotic or pathogenic microorganisms, and these glycosylated isoflavones are often one of the constitutive storage forms of isoflavones that accumulate in the central vacuole or other compartments. Hence, the content of glycosylated isoflavones (isoflavone glycoside) is higher than that of their aglycone forms in some plants. This modification alters their solubility and chemical properties, leading to different bioavailability, pharmacokinetics, and biological activity toward human health relative to aglycone forms. UDP-glycosyltransferases (UGTs) catalyze the addition of the glycosyl group from a nucleotide sugar (such as UDP sugar donors) to an isoflavone aglycone. Hence, the two core isoflavones (genistein and daidzein) and two 4′-methoxy derivative isoflavones (biochanin A and formononetin) undergo 7-*O*-glycosylation by UGTs and produce isoflavone 7-*O*-glycosides, such as genistin, daidzin, astroside, and ononin. Glycitein also undergoes 7-*O*-glycosylation by UGTs to form isoflavone 7-*O*-glycoside glycitin. Malonylation follows the process of glycosylation, potentially preventing the enzymatic degradation of the glycosylated isoflavones through altering their lipophilicity. Hence, malonyl isoflavones are also among the constitutively storage forms of isoflavones. Genistin, daidzin, and glycitin are further converted into their respective malonyl derivatives—malonyl genistin, malonyl daidzin, and malonyl glycitin by malonyltransferases (MTs) [59,60,63].

When plants are exposed to some stress factors, such as pathogens, UV radiation and high metal salts, pterocarpans are synthesized from core isoflavones such as daidzein, formononetin, and genistein. These above isoflavones are first converted into 2′-hydroxyisoflavones by isoflavone 2′-hydroxylase (I2′H), and then catalyzed by a series of various enzymes to form various pterocarpans in various plants. For example, daidzein is finally metabolized into glyceollin I, II, and III through the formation of intermediate metabolite 2′-hydroxydaidzein in soybean. Formononetin is finally metabolized into medicarpin, glycinol, and pisatin through the formation of intermediate metabolite 2′-hydroxyformononetin in *Medicago truncatula* Gaertn. [64].

## 3. The Metabolism of Isoflavones

These simple O-substituted isoflavones are widely present in leguminous plants (such as soybean and clover), and these plants are used in large quantities as feed for farm animals; accordingly, the metabolic processes of isoflavones have been extensively studied in sheep, cattle, goats, poultry, and humans [65,66]. The previous section described that simple O-substituted isoflavones mostly exist in plants in their glycoside form. However, isoflavone glycosides usually have higher hydrophilicity, resulting in low absorbance and low bioavailability because of the additional sugar group in the structure. Numerous studies have pointed out that deglycosylation via intestinal β-glycosidase is an important first step in the processes of metabolism, excretion, and biological activities of the isoflavone glycosides (genistin, daidzin, astroside, ononin, and glycitin) of these simple O-substituted isoflavones (genistein, daidzein, biochanin A, formononetin, and glyceitin). When these isoflavone glycosides are ingested from the dietary supplements, they are converted to the free O-substituted isoflavone aglycones through the hydrolysis of glycosidic bonds (deglycosylation or hydrolysis reaction) by intestinal β-glycosidase of the small intestinal microbiota and gastrointestinal mucosal cells. Then, the released aglycones are either absorbed intact through the epithelium of the small intestine (jejunum) or subjected to further metabolic processes (such as glucuronidation and sulfanation) by the intestinal microbiota [67,68]. For example, daidzin, genistin, glycitin, astroside, ononin, and glycitin are decomposed into daidzein, genistein, glycitein, biochanin A, formononetin, and glyceitin by intestinal β-glycosidase, respectively (Figure 4A,B). These free aglycones are derived from the original dietary supplements and from glycoside hydrolysis, thus, they are absorbed at different times (two stages) after ingestion. In pharmacokinetic investigation, the time of peak plasma concentrations of aglycones after ingesting aglycones is from 4–7 h, but the time after ingesting the glycosides is shifted to between 8 and 11 h [68,69,70]. After the ingestion of the dietary supplements containing O-substituted isoflavones, the plasma concentration curve of aglycones shows a double-peak pattern. There is an early increase in aglycone plasma concentrations within 30 min and 2 h after ingestion and then a second peak appears after between 4a and 8 h [71]. Hence, O-substituted isoflavone aglycones, which are less abundant, are first absorbed in the stomach, duodenum, and proximal jejunum. Then, the initial hydrolysis (deglycosylation) of O-substituted isoflavone glycosides is the rate-limiting step for absorption of these glycosides. Once absorbed, the free aglycones are transported to the liver, and then the hepatic microsomes catalyze the conjugation of O-substituted isoflavone glycosides with glucuronate and sulfate, such as daidzein, genistein, and glycitein at the 7- or 4′-position, and biochanin A and formononetin at the 7-position. These glucuronides are excreted in the bile and undergo enterohepatic circulation [72]. Immediately after, a small amount of unhydrolyzed and unabsorbed isoflavone glycosides in the small intestine is decomposed by enterobacterial enzymes in the colon along with the sulfated and glucuronidated forms of isoflavones, which are excreted into the small intestine through enterohepatic circulation. In fact, isoflavone glucuronides (~75%) are the main metabolites of O-substituted isoflavones in the plasma, followed by isoflavone sulfates (~24%), while aglycones only account for less than 1% [69]. Finally, these isoflavone glucuronides and sulfates are excreted in the urine due to their higher water solubility.

In addition, 7-hydroxyisoflavones (genistein and daidzein) are converted by other metabolic processes (such as reduction and hydroxylation) by the intestinal microbiota and liver microsomes. Genistein is first reduced to dihydrogenistein, and then metabolized to 6′-hydroxy-*O*-DMA, 4-hydroxyphenyl-2-propionic acid (HPPA), and *p*-ethyl-phenol through hydroxylation (Figure 4A). Daidzein is also first reduced to dihydrodaidzein and then metabolized to O-DMA, S-equol, 3-hydroxy-equol, and 6-hydroxy-equol through hydroxylation at the 3′- or 6-position (Figure 4A). HPPA and 6′-hydroxy-*O*-DMA are the two main end metabolites of genistein in humans. O-DMA and S-equol are the two main end metabolites of daidzein in humans [73,74]. Then, S-equol is also conjugated with glucuronate and sulfate at the 7- or 4′-position. Simple 7-hydroxyisoflavones (genistin, genistein, daidzin, daidzein, S-equol, and O-DMA) are mainly excreted with isoflavone glucuronides and sulfates in the urine, whereas a small amount of 7-hydroxyisoflavones is eliminated in the feces.

However, 4′-methoxyisoflavones (biochanin A and formononetin) are firstly converted to daidzein and genistein through O-demethylation. After demethylation, biochanin A (formononetin) and its metabolite genistein (daidzein) undergo glucuronidation and sulfanation. Furthermore, its metabolite genistein (daidzein) is further reduced and hydroxylated to HPPA and 6′-hydroxy-*O*-DMA (O-DMA and equol) by liver cytochrome P450 isoforms. On the other hand, biochanin A (formononetin) is hydroxylated to 3′,5,7-trihydroxy-4′-methoxyisoflavone (3′,7-dihydroxy-4′-methoxyisoflavone), 5,6,7-trihydroxy-4′-methoxyisoflavone (6,7-dihydroxy-4′-methoxyisoflavone), and 5,7,8-trihydroxy-4′-methoxyisoflavone (7,8-dihydroxy-4′-methoxyisoflavone) at the 3′-,6-, or 8-position by hepatic microsomes. Then, these hydroxylate metabolites are demethylated to additional metabolites [75,76,77] (Figure 4A).

As for glycitein, only a small amount is converted to daidzein through demethoxylation because the proximity of the 6-methoxyl and the 7-hydroxyl groups blocks the demethoxylation. It is predominately reduced to dihydroglycitein, and then metabolized to 6-methoxy equol, 5′-methoxyl-*O*-DMA, and dihydro-6,7,4′-trihydroxyisoflavone by liver cytochrome P450 isoforms [78] (Figure 4B).

## 4. The Protective Mechanism of Genistein and Genistin against Cerebral IR

Genistein (Figure 2) is a member of the 7-hydroxyisoflavone class and possesses a basic structural skeleton of isoflavones as a precursor of other isoflavones. It mainly exists in the Fabaceae family, such as *Glycine max* (L.) Merr. (soybean), *Medicago* spp. (alfalfa), *Psoralea corylifolia* L., *Pueraria* spp., and *Trifolium* spp. (clover). In addition to genistein, these above-mentioned leguminous plants contain the glycosylated, malonylated, and acetylated derivatives of genistein, such as genistin, malonyl genistin, and acetyl genistin. We searched and identified the relevant studies from 1953 to 2022 in PubMed, Web of Science, and SDOL databases to write this review. For data mining, the following MeSH words were used in the above databases: “phytoestrogen”, “isoflavone”, ”genistein”, “genistin”, “glutamate”, “NMDA”, “hypoxia”, “oxygen deprivation”, “stroke”, “ischemia or ischemic”, “reperfusion”, and “ischemia/reperfusion”. To date (31 July 2022), there are approximately 42 studies about the protective effects of genistein against cerebral IR, including two review articles. However, there are no studies about genistin against cerebral IR. Therefore, this section organizes the important results of the 42 articles about genistein against cerebral IR.

### 4.1. The Protective Effects of Genistein In Vitro Models

Genistein (10–100 μM) decreased the release of glutamate from hippocampal synaptosomes evoked by KCl-depolarization in a concentration-dependent manner. Genistein also decreased both the basal [Ca^2+^]i and the initial [Ca^2+^]i increase evoked by KCl-depolarization. When various voltage-gated calcium channel (VDCC) inhibitors and genistein were given simultaneously, only P/Q-type VDCC inhibitor ω-agatoxin IVA could increase the above effects of genistein [79]. Kajta′s report pointed out that genistein (10–10,000 nM) reduced the glutamate (1 mM)-induced apoptosis and the increase in caspase-3 activity in primary mouse hippocampal, neocortical and cerebellar cells. ICI 182,780 (ER antagonist) could block the preventive effects of genistein in primary mouse hippocampal cells. However, SB 216763 (glycogen synthase kinase 3β (GSK-3β) inhibitor) potentiated the preventive effects of genistein. Aryl hydrocarbon receptor antagonist-α-naphthoflavone exhibited a biphasic action: it enhanced the preventive effects of genistein toward a short-term exposure (3 h) to glutamate but antagonized the preventive effects of genistein toward prolonged exposure (24 h) [80]. In a primary cortical cell experiment, H_2_O_2_ (500 μM) was used as an oxidative stress inducer. Genistein (0.01–1 μM, pretreatment for 24 h) attenuated ROS generation, caspase-9 and caspase-3 activities, and neuronal apoptosis. Genistein could reverse the ratio of B-cell lymphoma 2 (Bcl-2) and Bcl-2-associated X protein (Bax), as well as suppress the phosphorylation of c-Jun NH2-terminal kinase (JNK), extracellular signal-regulated kinase (ERK), nuclear factor kappa-light-chain-enhancer of activated B cells (NF-κB) p65 subunit and inhibitor kappa B (IκB) [81]. Another report pointed out that genistein (10–500 nM) protected primary cortical neurons (only expressed ERβ) from thapsigargin (50 nM for 48 h, an endoplasmic reticulum calcium-ATPase inhibitor)-induced apoptosis. Genistein reduced the number of apoptotic neurons and caspase-3 activation. This effect was blocked by ICI 182,780 [82]. In the OGD series study of Liu et al., genistein (1 mM) decreased the apoptotic neuronal cell death in primary neurons and PC12 cells. Genistein further restored the voltage-activated potassium and sodium currents, as well as the expression of α-amino-3-hydroxy-5-methyl-4-isoxazolepropionic acid receptor (AMPA) GluR2 subunit and NMDA receptor 2 (NR2) in OGD neuronal cells [83,84]. Furthermore, genistein partially retained the cell viability and poly (ADP-ribose) polymerase 1 (PARP-1) cleavage; however, it did not decrease the apoptosis and caspase-3 activities after recurrent ischemic (OGD/reoxygenation (OGD/R)) injury in HT22 neuronal cells. Genistein only increased the expression of cytochrome c oxidase subunit 1 (CO-1) but did not alter the generation of ROS and the expression of hypoxia-inducible factor 1α (HIF-1α) and glucose transporter 1 (GLUT1) [85]. Hence, genistein possessed protective and anti-apoptotic effects against H_2_O_2_, thapsigargin and OGD through decreasing glutamate release by inhibiting VDCC and modulating voltage-dependent ion currents, glutamate receptor signaling, the ERβ/GSK-3β signaling pathway, the Bcl-2-related antiapoptotic pathway, and the mitogen-activated protein kinase (MAPK)/IκB/NFκB pathway. Moreover, Huang et al. pointed out that genistein directly inhibited NMDA receptor function to block the NMDA-activated current in a hippocampal slice culture and HEK293 cells transiently expressing rat recombinant NMDA receptors [86]. However, the aryl hydrocarbon receptor played a biphasic role in the protective effects of genistein against glutamate-induced excitotoxicity.

### 4.2. The Protective Effects of Geinistein In Vivo Cerebral IR Models

In a singlet oxygen-induced cerebral stroke model established with photoactivated Rose Bengal dye (intravenous administration) and transcranial green light illumination, genistein (16 mg/kg, treated every 6 h from 24 h prior to irradiation until 24 h after irradiation) decreased the cerebral lesion in balb/c mice [87]. Rumman et al. used a designated chamber flushed with 10% oxygen to construct a hypoxia model. Genistein (20–30 mg/kg, oral administration for 28 days) ameliorated the hypoxia-induced cognitive dysfunctions and reduced the hippocampal oxidative stress state. Genistein decreased the expression of pro-inflammatory cytokines (TNF-α, IL-1β, IL-6, and monocyte chemoattractant protein-1 (MCP-1)) and increased the expression of anti-inflammatory cytokines (IL-10) in the hippocampus. Genistein further enhanced the expression of brain-derived neurotrophic factor (BDNF), cAMP response element-binding protein (CREB), CREB-binding protein (CBP), and insulin-like growth factor 1 (IGF-1) in the hippocampus [88]. In another hypoxia model constructed with right common carotid artery ligation in neonatal mice, genistein (10 mg/kg, intraperitoneal administration once daily for 3 consecutive days before and once immediately after ligation) attenuated neuronal apoptosis and promoted the long-term recovery of brain atrophy and neurological function. Genistein reduced oxidative stress and neuroinflammation state. Meanwhile, genistein increased the ratio of Bcl-2/Bax, the HO-1 expression, and the nuclear translocation of Nrf-2, but inhibited cleaved caspase-3 expression and the phosphorylation of NF-κB and IκB [89]. In the 4-VO ischemia model (permanent bilateral vertebral artery occlusion and 15-min bilateral carotid artery occlusion), signal transducer and activator of transcription 3 (STAT3) phosphorylation and STAT3 DNA binding activity increased with the reperfusion time until 72 h after reperfusion. Genistein (5–20 mg/kg, intraperitoneal administration at 20 min before ischemia) could prevent the above changes at 72 h of reperfusion in a dose-dependent manner [90]. Another report also pointed out that genistein (15 mg/kg, intraperitoneal administration at 0 min before and again at 24 h after ischemia) attenuated the apoptotic neuronal cell death in the hippocampus of 4-VO ischemia rats. Genistein further decreased ROS generation, malondialdehyde (MDA) concentration, cytochrome c levels, and caspase-3 activation [91]. In the same 4-VO ischemic experiment, genistein (1.0 mg/kg, intravenous administration at 5 min after ischemia) attenuated the apoptotic neuronal cell death in the hippocampal CA1 region and improved the spatial memory in the Morris water maze. Genistein further increased endothelial nitric oxide synthase (eNOS) phosphorylation, Kelch-like ECH-associated protein 1 (Keap1) S-nitrosylation, nuclear translocation of nuclear factor erythroid 2-related factor 2 (Nrf-2) and heme oxygenase-1 (HO-1) expression, as well as decreased the levels of 8-hydroxy-2-deoxyguanosine (8-OHdG) and 4-hydroxynonenal (4-HNE) in the hippocampal CA1 region. However, e-NOS inhibitor L-N^G^-nitro arginine methyl ester (L-NAME) could block these above effects of genistein [92]. As for global cerebral IR model in gerbils, genistein (3–10 mg/kg, intraperitoneal administration at 5 min after ischemia) increased the survival of pyramidal cells in the hippocampal CA1 region and decreased MDA concentrations in the brain. Genistein also reversed IR-induced memory impairment and hyperlocomotion and antagonized the decrease in the total spectral power of electroencephalography. Then, 4-[2-phenyl-5,7-bis (trifluoromethyl) pyrazolo [1, 5-a] pyrimidin-3-yl] phenol (PHTPP) (a selective ERβ antagonist) blocked the protective effects of genistein [93]. Rajput et al. found that genistein (2.5–10 mg/kg, intraperitoneal administration for 2 weeks before ischemia) could decrease the infarct size and neuronal apoptosis in global cerebral IR with streptozotocin (STZ)-induced diabetic mice. Genistein could counteract the cognitive impairment and motor dysfunction induced by global cerebral IR in STZ-induced diabetic mice. Genistein further reduced the cerebral oxidative stress profile and circulating dipeptidyl peptidase-4 (DPP-4) activity but elevated the glucagon-like peptide-1 (GLP-1) concentration [94]. In the commonly used IR–MCAO model, genistein (2.5–10 mg/kg, oral administration for 2 weeks before MCAO) reduced the infarct volume and cell apoptosis, as well as improved the neurological deficit after ischemia. Genistein also decreased ROS production and MDA concentration, as well as increased the activities of antioxidant enzymes such as superoxide dismutase (SOD) and glutathione peroxidase (GSH-Px) in the cortex of transient MCAO mice (occlusion for 60 min). Genistein further decreased mitochondria ROS levels, cytochrome c release and caspase-3 activation. In addition, genistein suppressed the phosphorylation of NF-κB p65 and IκB [95]. In a rat transient MCAO experiment (occlusion for 90 min), genistein (500 ppm, intragastric treatment prior to ischemia for 2 weeks) reduced infarct size and neurological deficits in ovariectomized rats. Genistein further decreased the expression of gp91^phox^ and superoxide levels in the brain of ovariectomized rats [96]. In another rat transient MCAO experiment, genistein (10 mg/kg, intraperitoneal administration at 5 min after ischemia) attenuated the brain edema, infarct volume and neuronal degeneration. Genistein further decreased the oxidative stress profiles (increased MDA levels and decreased nuclear respiratory factor 1 (NRF-1) levels) and the activities of caspase-9 and caspase-3 [97]. Genistein (10 mg/kg, subcutaneous administration at 30 min after ischemia) also attenuated the neurological deficits and further decreased plasma thromboxane A_2_ (TXA_2_) concentration and leukocyte–platelet aggregates in transient MCAO (occlusion for 90 min) rats. Genistein could restore the contractile response of the MCAO rat carotid artery to U-46619 (TXA_2_ analogue) and the releasate from collagen-activated platelets [98]. Later, Wang′s recent series of studies on genistein for transient MCAO models, genistein (10 mg/kg, intraperitoneal administration for 2 weeks before ischemia) also alleviated the infarct volume, apoptotic cell death and neurological deficits in MCAO ovariectomized rat. Genistein could increase the ratio of Bcl-2/Bax and the phosphorylation of ERK 1/2. Then, U0126 (ERK 1/2 inhibitor) could augment these above effects of genistein [99]. Genistein further reduced the inflammatory factor release (tumor necrosis factor-α (TNF-α), interleukin-1β (IL-1β), IL-18 and IL-6) and the expression of microglial NOD-like receptor protein 3 (NLRP3) inflammasome-related proteins (pro–caspase-1, cleaved–caspase-1 and NLRP3) in reproductively transient MCAO senescent mice (17–18 months old) [100]. Genistein (10 mg/kg, intraperitoneal administration at 6 h, 24 h, and 48 h after reperfusion) reversed the expression of neuronal G protein-coupled estrogen receptor (GPER) and peroxisome proliferator-activated receptor-gamma coactivator 1α (PGC-1α), and decreased the interaction of NLRP3 with apoptosis-associated speck-like protein containing a CARD (ASC) in transient MCAO ovariectomized rat. Then, neuron-specific GPER or PGC-1α knockdown using AAVs could block these above effects [101]. In another experiment performed by Miao et al., genistein further increased the expression of Nrf-2 and NAD (P)H: quinone oxidoreductase-1 (NQO-1), as well as reduced ROS in the ischemic brain region of transient MCAO ovariectomized rats [102]. Lu′s report pointed out that genistein increased the expression of phosphatidylinositol 3-kinase (PI3K) and the phosphorylation of protein kinase B (Akt) and mammalian target of rapamycin (mTOR) in the ischemic brain region of transient MCAO ovariectomized rats (occlusion for 90 min) [103]. When MCAO ovariectomized rats were subcutaneously administered with a low-dose (0.1 mg/kg) of genistein using an Alzet osmotic mini-pump for 2 weeks, the number of surviving neurons in the hippocampal CA1 region was increased. However, the elevation in plasma corticosterone levels, total glucocorticoid receptor (GR) proteins and GR nuclear translocation in the hippocampal CA1 region of transient MCAO ovariectomized rats was decreased by treatment with genistein. Furthermore, genistein increased the GR–murine double minute 2 (Mdm2) interaction and the ubiquitination level of GR protein [104]. Hence, genistein can be considered as a potential therapeutic agent for cerebral IR. Its protective mechanism against cerebral IR is related to antioxidant activities via modulating the eNOS/Keap1/Nrf-2 signaling pathway and NADPH oxidase, anti-inflammatory activities via modulating the PI3K/AKT/mTOR signaling pathway, GPER/PGCL-1α or ERK/NLRP3/ASC signaling pathway and IκB/NFκ-B signaling pathway, and anti-apoptotic activities via the Bcl-2-regulated anti-apoptotic pathway. Its effects were also mediated by the neuronal survival BNDF or IGF-1/CREB pathway. On the hand, genistein can also exert protective effects against cerebral IR through decreasing the stress of GR signaling via ubiquitination and proteasome-mediated degradation of GR protein, as well as protecting vascular reactivity by decreasing TXA_2_ and aggregates production. On the basis of these studies, the detailed protective mechanism of genistein against cerebral IR is shown in Figure 5.

## 5. The Protective Mechanism of Daidzein and Daidzin against Cerebral IR

Daidzein (Figure 2) is also a member of the 7-hydroxyisoflavone class and possesses a basic structural skeleton of isoflavones as a precursor of other isoflavones. It mainly exists in the Fabaceae family, along with genistein and glycitein. In addition to daidzein, these above-mentioned leguminous plants contain the glycosylated, malonylated and acetylated derivatives, such as daidzin, malonyl daidzin and acetyl daidzin. In general, the content of daidzein and its derivates in the total isoflavones of soybeans is nearly equal to that of genistein and genistein derivatives. However, daidzin and malonyl daidzin are often richer than daidzein in soybean. We searched and identified the relevant studies from 1953 to 2022 in PubMed, Web of Science and SDOL databases to write this review. For data mining, the following MeSH words were used in the above databases: “phytoestrogen”, “isoflavone”, ”daidzein”, “daidzin”, “glutamate”, “NMDA”, “hypoxia”, “oxygen deprivation”, “stroke”, “ischemia or ischemic”, “reperfusion”, and “ischemia/reperfusion”. To date (31 July 2022), there are 16 studies about the protective effects of daidzein against cerebral IR in the PubMed, Web of Science, and SDOL databases. However, there are no studies about daidzin against cerebral IR. Therefore, this section organizes the important results of the 16 articles about daidzein against cerebral IR.

### 5.1. The Protective Effects of Daidzein In Vitro Models

In primary cortical cells exposed to glutamate (300 μM), thapsigargin (a calcium-ATPase inhibitor) (50 nM), hypoxia, or OGD, pretreatment with daidzein (1 μM) inhibited the above aspects of ischemic cell death. Daidzein (1 μM) further decreased the expression of cleaved capspase-3 and the cleavage of α-spectrin. ICI 182,780 (an ER antagonist) or LY294008 (a PI3K inhibitor) blocked these effects of daidzein [105]. Another report pointed out that daidzein (0.05–5 μM) decreased cell death in a primary culture of rat cortical neurons exposed to OGD. Daidzein (5 μM) increased synaptic vesicle recycling at nerve terminals in a primary culture of rat cerebellar granule cells. These effects were inhibited by a peroxisome proliferator-activated receptor γ (PPARγ) antagonist T0070907 (1 μM). In addition, daidzein increased PPARγ transcriptional activity and PPARγ nuclear protein levels but decreased PPARγ cytosolic protein levels. These effects were not due to binding to the receptor ligand site according to a TR-FRET PPARγ competitive binding assay [106]. Daidzein (0.1–10 μM) reduced the glutamate-induced apoptosis and caspase-3 activities in a primary culture of rat hippocampal, neocortical, and cerebellar cells in a similar manner. Daidzein (0.1–1 μM) alleviated the glutamate-induced loss of membrane mitochondrial potential in a primary culture of rat hippocampal cells. A selective ERβ antagonist, PHTPP, and a selective G-protein-coupled receptor 30 (GPR30) antagonist, G15, blocked these effects of daidzein on glutamate-induced excitotoxicity. In siRNA ERβ- and siRNA GPR30-transfected hippocampal neuronal cells, daidzein did not inhibit glutamate-induced excitotoxicity [107]. Therefore, these results demonstrated that daidzein possessed protective properties against various aspects of ischemia through activating ERβ and GPR30 intracellular signaling pathways and increasing PPARγ activity via post-translational modifications. On the other hand, daidzein (5–20 μM) decreased the production of NO and inhibited the expression of TLR4, iNOS, and COX-2 in LPS-treated BV2 microglial cells [108]. In addition, daidzein produced the vascular relaxation response in a rabbit basilar artery and this effect was not blocked by L-NAME, 1H- [1,2,4] oxadiazolo [4, 3-a] quinoxalin-1-one (selective inhibitor of NO-sensitive guanylyl cyclase), 4H-8-bromo-1,2,4-oxadiazolo (3, 4-d) benz (b) (1,4) oxazin-1-one (specific soluble guanylyl cyclase inhibitor), or indomethacin [109]. In cultured hippocampal neurons, daidzein promoted axonal outgrowths through upregulating the ERβ/PKCα/growth-associated protein 43 (GAP-43) signaling pathway [110] and neuronal cell proliferation via the BDNF/tyrosine kinase receptor (Trk) pathway [111].

### 5.2. The Protective Effects of Daidzein In Vivo Cerebral IR Models

In a rat transient MCAO model, daidzein (10 mg/kg, single intraperitoneal administration at 5 min after MCAO) alleviated neuronal damage in the ischemic brain, as well as neurological dysfunction. Daidzein increased SOD activity and expression but decreased MDA levels in the ischemic brain. Daidzein further decreased the elevated immunoreactivity of caspase-3 and caspase-9 in the ischemic brain tissue [112]. In another transient MCAO experiment (occlusion for 2 h), daidzein (10–30 mg/kg, intraperitoneal administration for 14 days) alleviated infarct volume, brain edema, and neurological deficits; meanwhile, it enhanced the phosphorylation of Akt, Bcl2-associated agonist of cell death (BAD), and mTOR, but reduced the cleaved caspase-3 level. Moreover, it enhanced the expression of BDNF and the phosphorylation of CREB. PI3K inhibitor LY294002 could abolish these above effects [113]. In a rat spinal cord ischemia reperfusion injury (SCIRI) model, daidzein (10 mg/kg, intraperitoneal administration for 14 days) alleviated neuronal apoptosis, locomotor activity, and edema in SCIRI rats. Daidzein also increased the activities of antioxidant enzymes (SOD and catalase), as well as decreased the levels of inflammatory markers (TNF-α and NF-κB p65) and the activities of myeloperoxidase (MPO) and caspase-3 in the injured spinal cord. Furthermore, daidzein increased the expression of PI3K and Bcl-2 and the phosphorylation of Akt, but decreased the expression of Bax [114]. Hence, daidzein attenuated IR-induced neuronal apoptosis and improved the neurological function through attenuating oxidative stress and the inflammatory response via upregulating the PI3K/Akt/mTOR signaling pathway and Bcl-2-regulated anti-apoptotic pathway. On the other hand, daidzein (0.2 mg/kg subcutaneous administration daily for 7 days) could modulate cerebral artery reactivity by enhancing the synthesis and release of endothelium-derived NO. This effect might be related to the altered expression of key proteins (calmodulin and caveolin) [115]. On the basis of these studies, the detailed protective mechanism of daidzein against cerebral IR is shown in Figure 6**.**

## 6. The Protective Mechanism of Biochanin A and Astroside against Cerebral IR

Biochanin A (Figure 2) is an O-methylated isoflavone biosynthesized from genistein, belonging to the 7-hydroxyisoflavone and 4′-methoxyisoflavone class. It was first isolated from the leaves and stems of *Trifolium pratense* L. (Fabaceae) and later reported to exist in other legumes, such as *Arachis hypogaea* L., *Astragalus* spp., *Cassia fistula* L., *Cicer arietinum* L., *Dalbergia sissoo* Roxb., *Glycine max* (L.) Merr., and *Medicago sativa* L. [54,55]. The content of biochanin A in the sprouts of *C. arietinum* is the highest (9.4 mg/g) among these legume plants and the most abundant among all isoflavones in the sprout. In the sprouts of *C. arietinum*, the 7-*O*-glycoside of biochanin A (astroside or sissotrin) was also found [116]. We searched and identified the relevant studies from 1953 to 2022 in PubMed, Web of Science, and SDOL databases to write this review. For data mining, the following MeSH words were used in the above databases: “phytoestrogen”, “isoflavone”, ”biochanin A”, “astroside”, “sissotrin”, “glutamate”, “NMDA”, “hypoxia”, “oxygen deprivation”, “stroke”, “ischemia or ischemic”, “reperfusion”, and “ischemia/reperfusion”. To date (31 July 2022), there are 12 studies about the protective effects of biochanin A against cerebral IR in the PubMed, Web of Science, and SDOL databases. However, there are no studies about astroside against cerebral IR. Therefore, this section organizes the important results of the 12 articles about biochanin A against cerebral IR.

### 6.1. The Protective Effects of Biochanin A In Vitro Models

In PC12 cells [117], pretreatment with biochanin A (1–100 μM) increased glutamate (10 mM)-induced apoptotic cell death in a concentration-dependent manner. The effects of biochanin A on glutamate-induced cytotoxicity might have been mediated by preventing glutathione (GSH) depletion and caspase-3 activation. In primary cortical cells and HT4 neuronal cells, pretreatment with biochanin A (25 μM) decreased glutamate-induced toxicity and apoptosis via inducing glutamate oxaloacetate transaminase (GOT) mRNA expression. This increase was blocked when ERRE sites were mutated [118]. On the other hand, biochanin A (1.25–5 μM) could decrease the production of ROS, RNS, and cytokines (prostaglandin E2 (PGE2), TNF-α, and IL-1β) induced by lipopolysaccharide (LPS) in BV2 microglial cells. It could inhibit the protein expressions of iNOS, cyclooxygenase-2 (COX-2), myeloid differentiation primary response 88 (MyD88), and Toll-like receptor 4 (TLR4). It further inhibited the LPS-induced phosphorylation of Akt, JNK, ERK, and p38, as well as the activation of NF-κB. These effects were abolished by GW9662 (a specific antagonist for PPARγ) [119,120,121]. In human umbilical vein endothelial cells, biochanin A (10–40 μM) could also decrease the production of TNF-α, the expression of vascular cell adhesion protein 1 (VCAM-1), intercellular adhesion molecule 1 (ICAM-1), and E-selectin, and the activation of NF-κB induced by LPS. These effects were also blocked by GW9662 [122]. In the rabbit basilar artery, biochanin A produced the vascular relaxation response and this effect was blocked by L-NAME [109].

### 6.2. The Protective Effects of Biochanin A In Vivo Cerebral IR Models

In a rat transient MCAO (occlusion for 2 h) model, biochanin A (10–40 mg/kg, intraperitoneal pretreatment for 14 days) decreased the infarct size and brain edema, as well as improved neurological deficit. Biochanin A suppressed the inflammatory processes (including increased MPO activities, cytokine levels, and mRNA expression). Biochanin A also decreased the lipid peroxidation and enhanced the activities of antioxidant enzymes (including SOD and GSH-Px). Furthermore, biochanin A suppressed the expression of C/EBP-homologous protein (CHOP), the phosphorylation of p38 and IκB, and NF-κB p65 translocation. However, biochanin A increased the expression of glucose regulated protein 78 (GRP78), the phosphorylation of HO-1, and Nrf-2 translocation [123,124,125]. In another transient MCAO experiment, biochanin A (5–10 mg/kg, intraperitoneal pretreatment for 4 weeks) attenuated stroke lesion volume and improved sensorimotor function. This protection was blocked by GOT knockdown [118]. In an experimental subarachnoid hemorrhage model, biochanin A (10–40 μg/μL, intracisternal injection at 1 h after the induction) attenuated brain edema and improved neurobehavioral dysfunction. Biochanin A further decreased microglial activation and cytokine levels via decreasing the expression of TLR2, TLR4, Toll-interleukin-1 receptor domain-containing adapter protein (TIRAP), and MyD88 protein. Biochanin A also decreased neuronal apoptosis via activating the Bcl-2 family [126]. Hence, biochanin A exerted protective effects against cerebral IR through inducing GOT gene expression, decreasing endoplasmic reticulum stress, activating the Bcl-2-regulated anti-apoptotic pathway and Nrf-2/HO-1 pathway, and inhibiting the TLRs/TIRAP/MyD88/NF-κB signaling pathway. On the basis of these studies, the detailed protective mechanism of biochanin A against cerebral IR is shown in Figure 7**.**

## 7. The Protective Mechanism of Formononetin and Ononin against Cerebral IR

Formononetin (Figure 2) is also an O-methylated isoflavone biosynthesized from daidzein, belonging to a membrane of the 7-hydroxyisoflavone and 4′-methoxyisoflavone class. It exists in the leaves and roots of some Fabaceae plants, such as *Trifolium pratense* L., *Astragalus* spp., *Glycyrrhiza glabra* L., and *Pueraria lobata* (Willd.) Ohwi. [57,127]. Ononin (7-*O*-glycoside of formononetin) is also found in the above plants, such as *Astragalus* spp., while it has also been isolated from the rhizome and root of other plants and other plants, such as *Smilax scobinicaulis* C. H. Wright, *Millettia nitida* var. *hirsutissima* Z. Wei., and *Tunisian Ononis angustissima* L. [128]. We searched and identified the relevant studies from 1953 to 2022 in PubMed, Web of Science, and SDOL databases to write this review. For data mining, the following MeSH words were used in the above databases: “phytoestrogen”, “isoflavone”, ”formononetin”, “ononin”, “glutamate”, “NMDA”, “hypoxia”, “oxygen deprivation”, “stroke”, “ischemia or ischemic”, “reperfusion”, and “ischemia/reperfusion”. To date (31 July 2022), there are 10 studies about the protective effects of formononetin against cerebral IR in the PubMed, Web of Science, and SDOL databases. There are no studies about ononin against cerebral IR. Therefore, this section organizes the important results of the 10 articles about formononetin against cerebral IR.

Formononetin (10 μM) decreased apoptotic neuronal death caused by NMDA (100 μM) in primary cortical neurons. Formononetin further increased the expression of Bcl-2 and pro-caspase-3, as well as decreased the expression of Bax and caspase-3 [129]. In a transient MCAO (occlusion for 2 h) model, formononetin (12.5–50 mg/kg, intraperitoneal pretreatment for 14 days) reduced the infarct size and brain edema. Formononetin improved the neurological deficit and increased the number of neuronal dendritic spines. Formononetin increased the mRNA and protein expression of ER-α and Bcl-2, as well as decreased the mRNA and protein expression of Bax. Formononetin further increased the phosphorylation of Akt, ERK, and Trk A and B, as well as the expression of nerve growth factor (NGF), BDNF, β tubulin-III, and GAP-43 [130,131]. In another transient MCAO experiment (occlusion for 60 min), formononetin (30 mg/kg) alleviated the cerebral infarction and the neurological deficit, as well as reduced the mRNA levels of IL-6 and IL-1β in rat brain tissue, the protein levels of NLRP3, ASC, cleaved caspase-1, and cleaved IL-1β, and the phosphorylation of Janus kinase 2 (JAK2) and STAT3 in the MCAO rat brain tissue [132]. Hence, the benefit mechanism of formononetin against cerebral IR might be related to the reduction in neuroinflammation via the JAK/STAT/inflammasome pathway, the decrease in neuronal apoptosis via the Bcl-2-regulated anti-apoptotic pathway and increase the neuronal differentiation and synaptic plasticity via the BDNF/Trk and PI3K/AKT/ERK pathways. On the basis of these studies, the detailed protective mechanism of formononetin against cerebral IR is shown in Figure 8.

## 8. The Protective Mechanism of Equol, O-DMA, and Other Metabolites against Cerebral IR

According to many pharmacokinetic and metabolic reports, isoflavone glucuronides and sulfates are the main metabolites of isoflavones in human plasma. As for the free aglycone metabolites of isoflavones, dihydrogenistein, dihydrodaidzein, and dihydroglycitein are the intermediate metabolites of genistein, daidzein, and glycitein in plasma. Genistein and daidzein are the intermediate metabolites of biochanin A and formononetin. The end-products of genistein and biochanin A are HPPA, 6′-hydroxy-*O*-DMA, and *p*-ethyl-phenol. The end-products of daidzein and formononetin are O-DMA, S-equol, 3-hydroxy-equol, and 6-hydroxy-equol. The end-products of glycitein are 6-methoxy equol and 5′-methoxyl-*O*-DMA. In addition, some hydroxylated compounds at the 3′-, 6-, or 8-position are also considered as end metabolites of these above isoflavones [73,74,75,76,77,78]. However, these metabolites do not necessarily have the pharmacological activities of these parent isoflavones. For example, isoflavone glucuronides, S-equol, 6-hydroxy-equol, and O-DMA also possessed ERβ activities, as did the parent isoflavones. However, the ERβ activities of isoflavone glucuronides, 6-hydroxy-equol, and O-DMA were weaker than their parent isoflavones [39,74]. The ERβ activity of S-equol is comparable to or even better than that of daidzein, while it is almost equivalent to that of genistein [133]. As for other metabolites, they may not be ERβ active (*p*-ethyl-phenol), or their activities have not been reported. Hence, the related literature on the pharmacological activity of S-equol, O-DMA, and HPPA was searched. We searched and identified the relevant studies from 1953 to 2022 in PubMed, Web of Science, and SDOL databases to write this review. For data mining, the following MeSH words were used in the above databases: “phytoestrogen”, “isoflavone”, ”equol”, “desmethylangolensin”, “ethyl-phenol”, “4-hydroxyphenyl-2-propionic acid”, “glutamate”, “NMDA”, “hypoxia”, “oxygen deprivation”, “stroke”, “ischemia or ischemic”, “reperfusion”, and “ischemia/reperfusion”. To date (31 July 2022), there are 10 studies about the protective effects of S-equol against cerebral IR in the PubMed, Web of Science, and SDOL databases. There are no studies about O-DMA and HPPA against cerebral IR.

In an in vitro hypoxia/reoxygenation injury model, equol (1–10 μM) dose-dependently restored the cell viability and decreased the ROS and MDA content in cortical cells and PC12 cells. Equol further decreased the protein expression of gp91^phox^ and the phosphorylation of Src. These above effects were blocked by Src siRNA but not gp91^phox^ siRNA [105,134,135]. On the other hand, equol (5–20 μM) decreased the production of NO and PGE2, as well as the secretion of cytokines (TNF-α and IL-6) in LPS-treated BV2 microglial cells. Equol further inhibited the expression of TLR4, iNOS, COX-2, IκB, and NF-κB, as well as the phosphorylation of MAPK induced by LPS in BV2 microglial cells. Additionally, equol protected N2a neurons from neuroinflammatory injury mediated by LPS-activated microglia through regulating the Bcl-2-related apoptotic pathway [108]. In the transient MCAO (occlusion for 2 h) model, equol (0.625–2.5 μg/kg, intragastric treatment prior to ischemia for 3 days) attenuated infarct volume, histological damage, and neurological deficits. Equol also decreased MDA levels, the protein expression of gp91^phox^, and the phosphorylation of Src in rat cortex. [135]. In another transient MCAO experiment (occlusion for 90 min), equol (250 ppm, intragastric treatment prior to ischemia for 2 weeks) reduced infarct size and neurological deficits in ovariectomized rats. Equol further decreased the expression of gp91^phox^ and superoxide levels in the brain of ovariectomized rats. In addition, equol reduced plasma MDA levels up to 7 days after injury [96]. Hence, the protective effect of equol might be mediated by the inhibition of Src-mediated cell signaling and the downregulation of ROS generating enzyme gp91^phox^.

## 9. Conclusions

Simple O-substituted isoflavones mainly include 7-hydroxyisoflavones, 7-hydroxy-4′-methoxyisoflavones, and their derivates, such as glycosylated and malonyl isoflavones. In plants, especially legumes, the content of glycosylated and malonyl isoflavones is higher than that of O-substituted isoflavone aglycones. After ingestion, the conjugated isoflavones are hydrolyzed to their aglycones by the intestinal microbiota, which are then absorbed in the small intestine and metabolized in the liver to glucuronides, sulfates, and other hydroxylated aglycones through glucuronidation, sulfation, reduction, hydroxylation, and ring-cleavage reaction. In earlier studies, scientists mainly focused on their ER properties due to their structural similarity to estrogen. According to the physiological role of ER and the pharmacological activity of estrogen, researchers also found that O-substituted isoflavones possessed multiple pharmacological activities, such as antioxidation, anti-inflammation, antimicrobials, anticancer, and neuroprotective properties. According to the bioavailability and affinity for ER, O-substituted isoflavone aglycones usually possess the better health benefits than their conjugates and metabolites. 5,7-Dihydroxyisoflavones have higher pharmacological potency than 7-hydroxyisoflavones. Collected from various databases, we also found more published works of genistein and biochanin on cerebral IR than those of daidzein, formononetin, and equol against cerebral IR. However, there are no studies about glycitein and isoflavone conjugates against cerebral IR. All relevant studies about O-substituted isoflavones and cerebral IR injury are listed in the table below (Table 1).

On the basis of the data of these published works, O-substituted isoflavones could serve as promising therapeutic compounds for the prevention and treatment of cerebral IR. The protective potential of O-substituted isoflavones against cerebral IR injury might be related to ER properties and neuron-modulatory, antioxidant, anti-inflammatory, and anti-apoptotic effects. Firstly, the neuron-modulatory mechanism of O-substituted isoflavones mainly include a decrease in glutamate signaling via inhibiting the VDCC and ERβ or GPR30/GSK-3β/GOT signaling pathway; as well as an increase in neuronal differentiation and synaptic plasticity via activating the BDNF/Trk and PI3K/AKT/ERK pathways. The antioxidant mechanism of O-substituted isoflavones might be related to the activation of the eNOS/Keap1/Nrf-2/HO-1 signaling pathway and NADPH oxidase, as well as the downregulation of Src-mediated cell signaling and gp91^phox^. The anti-inflammatory mechanism of O-substituted isoflavones might be related to the upregulation of ERβ or the GPR30/PI3K/AKT/mTOR signaling pathway and PPARγ activity; as well as to the downregulation of the TLRs/TIRAP/MyD88/NFκ-B signaling pathway. Lastly, the anti-apoptotic mechanism of O-substituted isoflavones is related to not only the above-mentioned antioxidant and anti-inflammatory activities, but also the Bcl-2-regulated anti-apoptotic pathway.

## Figures and Tables

**Figure 1 ijms-23-10394-f001:**
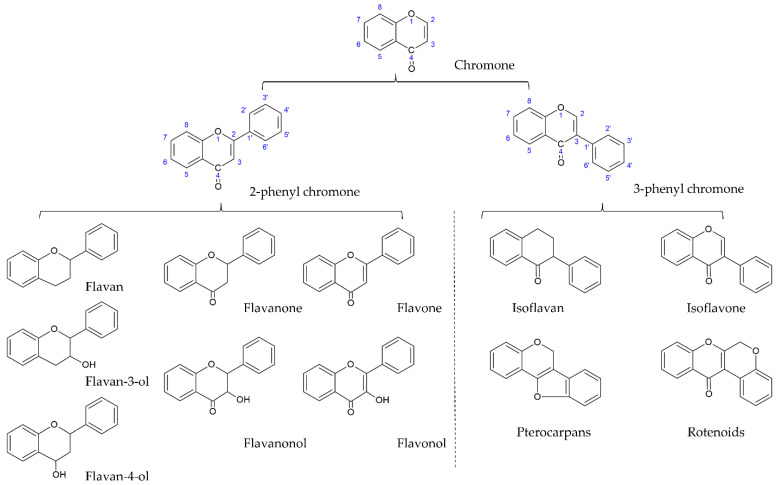
Core structures of flavonoids and isoflavonoids.

**Figure 2 ijms-23-10394-f002:**
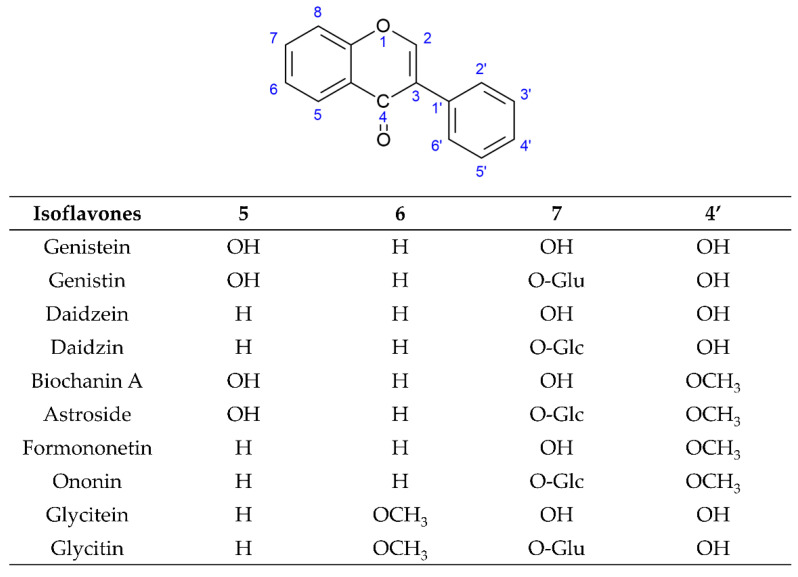
Structures of novel simple O-substituted isoflavones.

**Figure 3 ijms-23-10394-f003:**
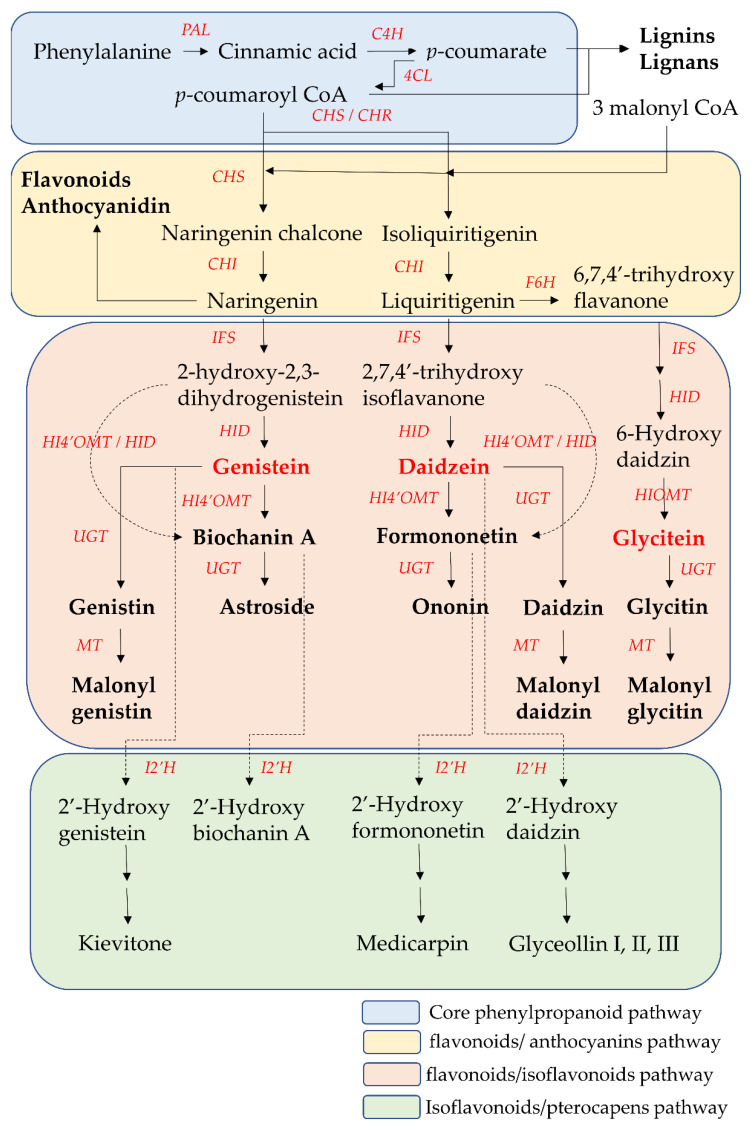
The core phenylpropanoid pathway and isoflavones biosynthesis pathway. 4CL, 4-coumarate CoA-ligase; C4H, cinnamate 4-hydroxylase; CHI, chalcone isomerase; CHR, chalcone reductase; CHS, chalcone synthase; F6H, flavonoid 6-hydroxylase; HI4′OMT, hydroxyisoflavanone 4′-specific O-methyltransferase; HID, 2-hydroxyisoflavanone dehydratase; HIOMT, hydroxyisoflavanone O-methyltransferase; I2′H, isoflavone 2′hydroxylase; IFS, isoflavone synthase; MT, malonyltransferases; PAL, phenylalanine ammonia-lyase. UGTs. UDP-glycosyltransferases.

**Figure 4 ijms-23-10394-f004:**
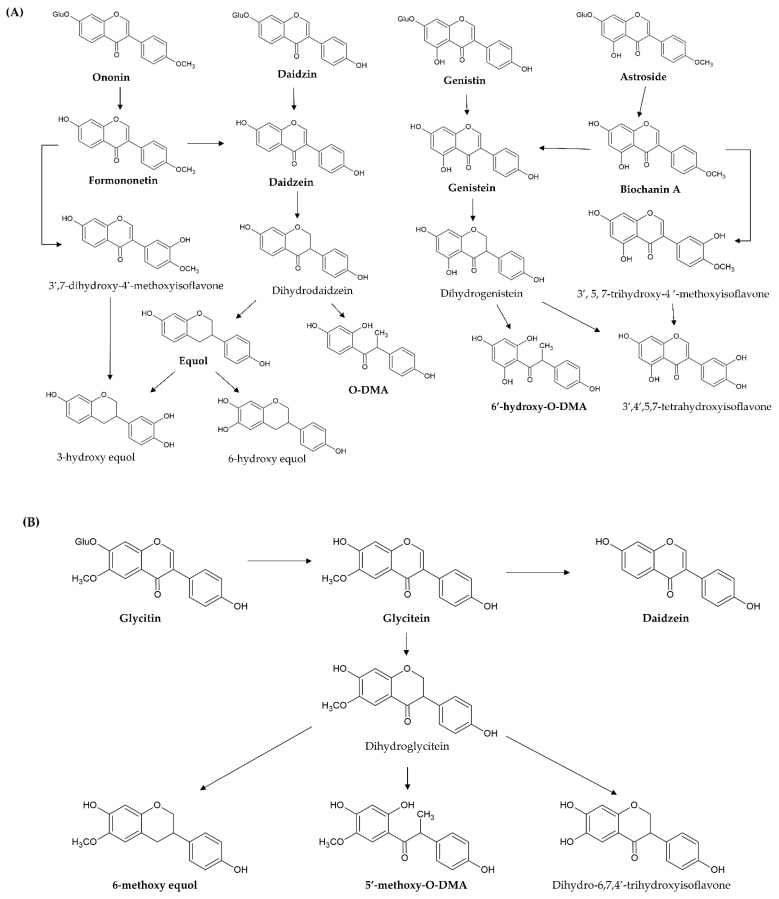
Proposed metabolic pathway of novel simple O-substituted isoflavones ((**A**) daidzin, daidzein, genistin, genistein, biochanin A, and formononetin; (**B**) glycitin and glycitein). O-DMA, O-desmethylangolensin.

**Figure 5 ijms-23-10394-f005:**
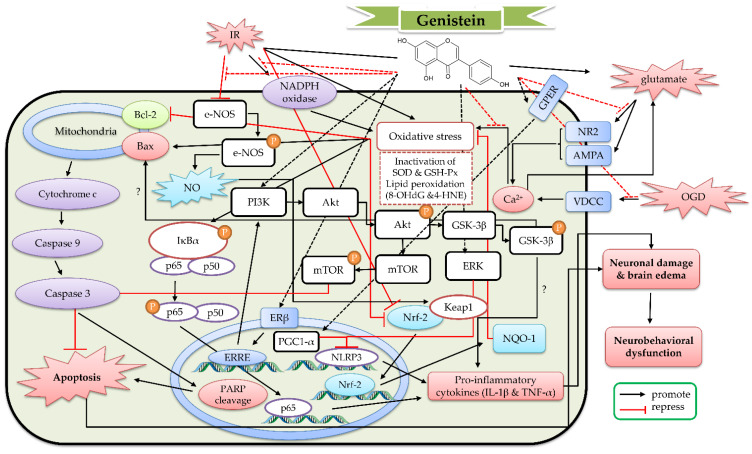
The protective effects of genistein and genistin against cerebral ischemia reperfusion (IR). 4-HNE, 4-hydroxynonenal; 8-OHdG, 8-hydroxy-2-deoxyguanosine; Akt, protein kinase B; AMPA, α-amino-3-hydroxy-5-methyl-4-isoxazolepropionic acid; Bax, Bcl-2-associated X protein; Bcl-2, B-cell lymphoma 2; eNOS, endothelial nitric oxide synthase; ERβ, estrogen receptor β; ERRE, estrogen-related receptor response element sites; GPER, G protein-coupled estrogen receptor; GSH-Px, glutathione peroxidase; GSK-3β, glycogen synthase kinase 3β; IκBα, Inhibitor kappa B; IL-1β, interleukin-1β; IR, ischemia reperfusion; Keap1, Kelch-like ECH-associated protein 1; mTOR, mammalian target of rapamycin; NADPH, nicotinamide adenine dinucleotide phosphate; NLRP3, NOD-like receptor protein 3; NO, nitric oxide; NR2, N-methyl-D-aspartate receptor 2; NQO-1, NAD (P)H: quinone oxidoreductase-1; Nrf-2, nuclear factor erythroid 2-related factor 2; OGD, oxygen and glucose deprivation; PARP, Poly (ADP-ribose) polymerase; PGC-1α, peroxisome proliferator-activated receptor-gamma coactivator 1α; PI3K, phosphatidylinositol 3-kinase; ROS, reactive oxygen species; SOD, superoxide dismutase; TNF-α, tumor necrosis factor α; VDCC, voltage-dependent calcium channel.

**Figure 6 ijms-23-10394-f006:**
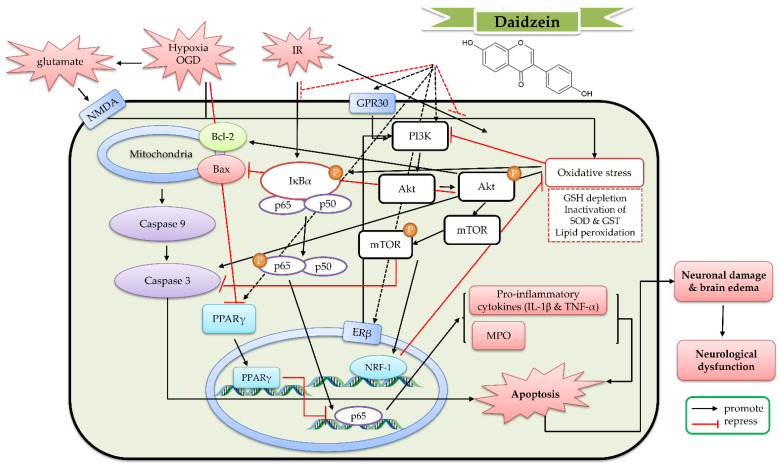
The protective effects of daidzein and daidzin against cerebral ischemia reperfusion (IR). Akt, protein kinase B; Bax, Bcl-2-associated X protein; Bcl-2, B-cell lymphoma 2, ERβ, estrogen receptor β; GPR30, G-protein-coupled receptor 30; GSH, glutathione; GST, glutathione sulfotransferase; IκB, Inhibitor kappa B; IL-1β, interleukin-1β; IR, ischemia reperfusion; MPO, myeloperoxidase; mTOR, mammalian target of rapamycin; NMDA, N-methyl-D-aspartate; NRF-1, nuclear respiratory factor 1; OGD, oxygen-glucose deprivation; PI3K, phosphoinositide 3-kinase; PPARγ, peroxisome proliferator-activated receptor γ; SOD, superoxide dismutase; TNF-α, tumor necrosis factor α.

**Figure 7 ijms-23-10394-f007:**
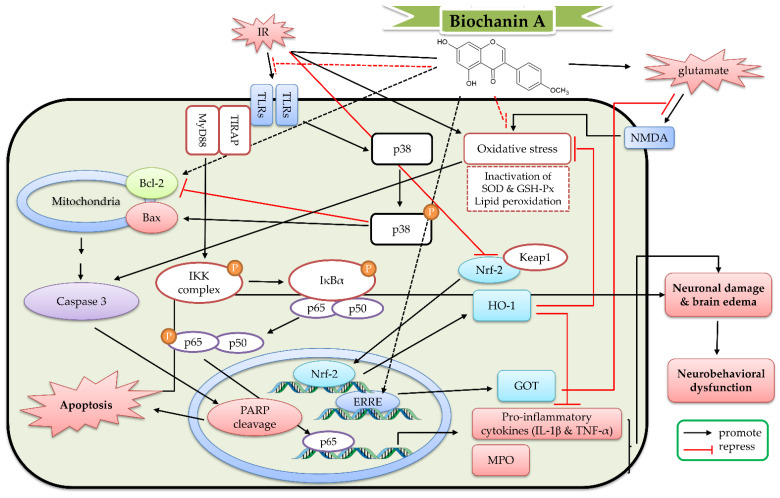
The protective effects of biochanin A against cerebral ischemia reperfusion (IR). Bax, Bcl-2-associated X protein; Bcl-2, B-cell lymphoma 2; ERRE, estrogen-related receptor response element sites; GOT, glutamate oxaloacetate transaminase; GSH-Px, glutathione peroxidase; HO-1, heme oxygenase-1; IκBα, Inhibitor kappa B α; IKK, IκB kinase; IL-1β, interleukin-1β; IR, ischemia reperfusion; Keap1, Kelch-like ECH-associated protein 1; MPO, myeloperoxidase; MyD88, myeloid differentiation primary response 88; NMDA, N-methyl-D-aspartate; Nrf-2, nuclear factor erythroid 2-related factor 2; PARP, Poly (ADP-ribose) polymerase; SOD, superoxide dismutase; TIRAP, Toll-interleukin-1 receptor domain-containing adapter protein; TLR, Toll-like receptor; TNF-α, tumor necrosis factor α.

**Figure 8 ijms-23-10394-f008:**
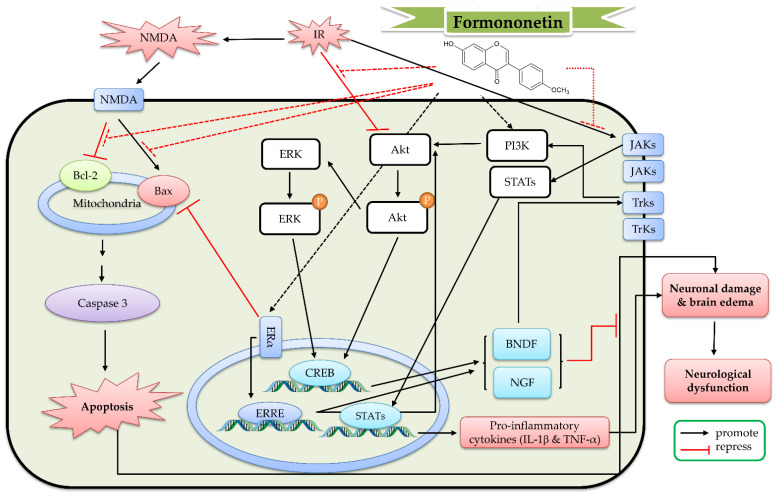
The protective effects of formononetin against cerebral ischemia reperfusion (IR). Akt, protein kinase B; Bax, Bcl-2-associated X protein; Bcl-2, B-cell lymphoma 2; BDNF, brain-derived neurotrophic factor; CREB, cAMP response element-binding protein; ERK, extracellular-signal-regulated kinase; ERRE, estrogen-related receptor response element sites; ERα, estrogen receptor α; IL-1β, interleukin-1β; JAKs, Janus kinases; NGF, nerve growth factor; NMDA, N-methyl-D-aspartate; PI3K, phosphoinositide 3-kinase; STATs, signal transducer and activator of transcriptions; TNF-α, tumor necrosis factor α, Trk, tyrosine kinase receptors.

**Table 1 ijms-23-10394-t001:** The protective effects of O-substituted isoflavones against ischemia reperfusion (IR) injury.

**1.** **In Vitro Experiments**							
O-Substituted Isoflavones	Applied Concentration	Application	Employed Cells	Employed Model	Observed Effects	Possible Mechanism	References
Genistein	0.1–1 μM	Pre-treatment for 24 h	primary cortical cells	Glutamate (300 μM) for 10 min Thapsigargin (50 nM) for 48 h Hypoxia for 16 h OGD for 2 or 5 h	↓ apoptosis	↑ ER, PI3K and MAPK	[105]
	10–10,000 nM	Co-treatment	Primary hippocampal, neocortical, and cerebellar cell	Glutamate (1 mM) for 3, 6, 24 h	↓ apoptosis	↓ aryl hydrocarbon receptor and ER/GSK3β	[80]
	10–500 nM	Co-treatment	primary cortical cells	Thapsigargin (50 nM) for 48 h	↓ apoptosis	ERβ	[82]
	0.01–1 μM	Pre-treatment for 24 h	primary cortical cells	H_2_O_2_ (500 μM) for 1 h	↓ apoptosis and neurotoxicity	↓ MAPK/ NF-κB pathway	[81]
	1 mM	Co-treament	Primary neurons	OGD for 3, 6, 24, 48 h	↓ apoptosis and neurotoxicity	↑ Kv currents ↓ Nav currents and glutamate pathway	[83]
	30 μM	Co-treament	PC12 cells	OGD for 1, 3, 6, 24 h	↓ apoptosis and neurotoxicity	↑ Kv currents ↓ glutamate pathway	[84]
	1 μM	Pretreatment for 1 h	HT22 cells	OGD/R for 18 h	↓ apoptosis and neurotoxicity	-	[85]
Daidzein	0.1–1 μM	Pre-treatment for 24 h	primary cortical cells	Glutamate (300 μM) for 10 min Thapsigargin (50 nM) for 48 h Hypoxia for 16 h OGD for 2 or 5 h	↓ apoptosis	↑ ER, PI3K	[105]
	0.05–5 μM	Pre-treatment for 24 h	primary cortical cells	OGD for 2 or 5 h	↓ apoptosis and neurotoxicity	↑ PPARγ	[106]
	1–10 μM	Co-treatment	primary neocortical, cerebellar, and hippocampal cell	Glutamate (1 mM) for 6 or 24 h	↓ apoptosis and neurotoxicity	↓ ERβ, GPR30 and caspase-3	[107]
Biochanin A	1–100 μM	Pre-treatment for 2 h	PC12 cells	Glutamate (10 mM) for 6 or 24 h	↓ apoptosis	↓ caspase-3	[117]
	25–50 μM	Pre-treatment for 24 h	HT4 neural cells	Glutamate (5 mM) for 24 h	↓ neurotoxicity ↓ inflammation	↑ GOT ↓ TLR4/MyD88/NF-κB, MAPK, and Akt	[118]
			pirmary cortical cells				
	2.5–100 μM	Co-treatment	BV2 cells	LPS (100 ng/mL) for 24 h	↓ inflammation	↑ PPARγ ↓ NF-κB	[119]
	5–20 μM	Pre-treatment for 1 h	BV2 cells	LPS (0.5 μg/mL) for 24 h	↓ inflammation	↓ MAPK	[120]
	1.25–5 μM	Co-treatment	BV2 cells	LPS (1 μg/mL) for 36 h	↓ inflammation and adhesion	↑ PPARγ ↓ NF-κB	[121]
	10–40 μM	Pre-treatment for 12 h	HUVEC cells	LPS (1 μg/mL) for 36 h	↓ neurotoxicity	↑ GRP78 ↓ CHOP and p38	[122]
	2–4 μM	Pre-treatment for 24 h	primary cortical cells	OGD for 2 h	↓ apoptosis	↓ caspase-3	[125]
Formononetin	1–10 μM	Pre-treatment for 12 h	primary cortical cells	NMDA (200 μM) plus glycine (10 μM) for 40 min	↓ excitotoxic injury	↑ Bcl-2	[129]
Equol	0.1–1 μM	Pre-treatment for 24 h	primary cortical cells	Hypoxia for 16 h OGD for 2 or 5 h	↓ apoptosis	-	[105]
	5–20 μM	Pre-treatment for 30 min	BV2, C6, and N2a cells	LPS (100 ng/mL) for 24 h	↓ neuroinflammation and activated microglia-induced neurotoxicity	↓ TLR4/JNK/NF-κB pathway	[108]
	0.1–10 μM	Pre-treatment for 1 h	PC12 cells	Hypoxia for 12 h	↓ cell death and ROS production	↓ gp91^phox^ and Src phosphorylation	[134,135]
**2.** **In Vivo Experiments**							
**O-Substituted Isoflavones**	**Applied dose**	**Application**	**Species**	**Employed Model**	**Observed Effects**	**Possible Mechanism**	**References**
Genistein	5–20 mg/kg, ip	Pre- and post-treatment	SD rats	4-VO	-	↓ STAT3 activation	[90]
	15 mg/kg, ip	Pre- and post-treatment	SD rats	4-VO	↓ hippocampal neuronal damage	↑ Bcl-2-related pathway	[91]
	3–10 mg/kg,	Post-treatment	Mongolian gerbils	common carotid occlusion	↓ neuronal damage and neurological deficits	ERβ	[93]
	500 ppm, po	Pre-treatment for 2 wk	SD rats (OVA)	MCAO for 90 min Followed by 24 h of reperfusion	↓ infarct size	↓ gp91^phox^	[96]
	2.5–10 mg/kg, po	Pre-treatment for 2 wk	C57/ BL6J mice	MCAO for 60 min	↓ infarct volume, neurological deficit, and apoptosis	↓ mitochondria-dependent apoptosis pathway and NF-κB	[95]
	1 mg/kg, iv	Post-treatment	SD rats	4-VO	↓ hippocampal neuronal damage, and neurological deficits	↑ eNOS/Nrf2/HO-1 pathway	[92]
	10 mg/kg, sc	Pre-treatment	SD rats	MCAO for 90 min	↓ infarct volume, neurological deficit, and vascular reactivity	↑ circulatory function	[98]
	10 mg/kg, ip	Pre-treatment for 2 wk	Kunming mice (OVA)	MCAO for 90 min	↓ infarct volume, neurological deficit, and apoptosis	↑ ERK activation	[99]
	0.1 mg/kg/day, sc	Pre-treatment for 2 wk	SD rats (OVA)	MCAO for 10 min	↓ hippocampal neuronal damage	↓ GR–Mdm2 interaction	[104]
	10 mg/kg, ip	Post-treatment	SD rats	MCAO for 5 min	↓ neuronal damage, brain edema, and neurological deficits	↑ NRF-1 ↓ mitochondria-mediated apoptotic pathway	[97]
	2.5–10 mg/kg, ip	Pre-treatment for 2 wk	STZ-induced Swiss mice	MCAO for 30 min	↓ infarct size, neurological deficits and glucose levels	↑ GLP-1 ↓ DPP-4	[94]
	10 mg/kg, ip	Pre-treatment for 2 wk	SD rats (OVA)	MCAO for 90 min	↓ infarct volume, neuronal damage, and neurological deficits	↑ Nrf-2/NQO-1 ↑ PI3K/Akt/mTOR pathway	[102,103]
	10 mg/kg, ip	Pre-treatment for 2 wk	senescent C57BL/6J mice	MCAO for 60 min	↓ infarct volume, neuronal damage, and neurological deficits	↓ NLRP3 Inflammasome	[100]
	10–30 mg/kg, po	Co-treatment for 2 wk	Swiss mice	Hypoxia for 2 wk	↓ cognition deficits	↑ IGF-1/CREB/BDNF	[88]
	10 mg/kg, ip	Pre-treatment for 3 days	neonatal C57BL/6 mice	MCAO for 60 min	↓ infarct volume, neuronal damage, and neurological deficits	↑ Nrf2/HO-1 pathway ↓ NF-κB pathway	[89]
	10 mg/kg, ip	Post-treatment	C57BL/6 mice (OVA)	MCAO for 10 min	↓ infarct volume, apoptosis, and neurological deficits	↑ GPER/PGC-1α pathway ↓ NLRP3 Inflammasome	[101]
Daidzein	10 mg/kg, ip	Post-treatment	SD rats	MCAO	↓ apoptosis, neuronal damage, and neurological deficits	↓ caspase-3	[112]
	20 mg/kg, ip	Pre-treatment for 1 wk	SD rats	Spinal cord IR for 25 min	↓ apoptosis, neuronal damage, and neurological deficits	↓ caspase-3 and NF-κB ↑ PI3K/Akt pathway	[114]
	10–30 mg/kg, ip	Post-treatment	ICR mice	MCAO for 2 h	↓ infarct volume, brain edema, and neurological deficits	↑ PI3K/Akt/mTOR pathway ↑ Akt/mTOR/BDNF pathway	[113]
Biochanin A	5–10 mg/kg, ip	Pre-treatment for 4 wk	C57BL/6 mice	MCAO for 60 min	↓ infarcted volume and sensorimotor deficit	↑ ERRE and GOT	[118]
	10–40 mg/kg, ip	Pre-treatment for 2 wk	SD rats	MCAO for 2 h Followed by 24 h of reperfusion	↓ infarct volume, brain edema, and neurological deficits	↑ Nrf-2/HO-1 pathway ↓ NF-κB	[123]
	10–40 mg/kg, ip	Pre-treatment for 2 wk	SD rats	MCAO for 2 h Followed by 24 h of reperfusion	↓ infarct volume, brain edema, and neurological deficits	↑ GRP78 ↓ CHOP and p38	[125]
	10–40 mg/kg, ip	Pre-treatment for 2 wk	SD rats	MCAO for 2 h	↓ infarct volume, brain edema, and neurological deficits	↓ p38 pathway	[124]
	10–40 μg/kg, icv	Post-treatment	SD rats	Subarachnoid Hemorrhage	↓ mortality, brain edema, and neurological deficits	↓ TLRs/TIRAP/MyD88/NF-κB pathway	[126]
Formononetin	12.5–25 mg/kg, ip	Pre-treatment for 2 wk	SD rats	MCAO for 2 h Followed by 24 h of reperfusion	↓ infarcted volume, brain edema, and neurological deficit	↑ PI3K/Akt and Bcl-2	[131]
	30 mg/kg	Post-treatment for 2 wk	SD rats	MCAO for 2 h Followed by 14 days of reperfusion	↑ injured neurological functions, neuronal differentiation, and synaptic plasticity.	↑ BDNF or NGF/Trk/PI3K/Akt/ERK	[130]
	30 mg/kg	Post-treatment for 3 d	SD rats	MCAO for 1 h Followed by 3 days of reperfusion	↓ neurological deficit and neuroinflammation	↓ JAK2/STAT3 pathway	[132]
Equol	250 ppm, po	Pre-treatment for 2 wk	SD rats	MCAO for 90 min Followed by 24 h of reperfusion	↓ infarct size	↓ gp91^phox^	[96]
	0.625–2.5 mg/kg, po	Pre-treatment for 3 d	SD rats	MCAO for 2 h Followed by 22 h of reperfusion	↓ mortality, neurological deficit, brain damage, and infarct volume	↓ gp91^phox^ and Src phosphorylation	[135]

4-VO: four-vessel occlusion; Akt, protein kinase B; Bcl-2: B-cell lymphoma 2; BDNF: brain-derived neurotrophic factor; CHOP: C/EBP-homologous protein; CREB: cAMP response element-binding protein; DPP-4: dipeptidyl peptidase-4; eNOS: endothelial nitric oxide synthase; ER: estrogen receptor; ERK: extracellular-signal-regulated kinase; ERRE: estrogen-related receptor response element sites; GLP-1: glucagon-like peptide-1; GOT: glutamate oxaloacetate transaminase; GPER: G protein-coupled estrogen receptor; GR: glucocorticoid receptors; GRP30: G-protein-coupled receptor 30; GSK-3β: glycogen synthase kinase-3β; HO-1: heme oxygenase-1; IGF-1: insulin-like growth factor 1; IR: ischemia reperfusion; JAK2: Janus kinase 2; LPS: lipopolysaccharides; MAPK: mitogen-activated protein kinase; MCAO: middle carotid artery occlusion; Mdm2: murine double minute 2; mTOR: mammalian target of rapamycin; NF-κB: nuclear Factor kappa-light-chain-enhancer of activated B cells; NGF: nerve growth factor; NLRP3: NOD-like receptor protein 3; NRF-1: nuclear respiratory factor 1; Nrf-2: nuclear factor erythroid 2-related factor 2; NMDA: N-methyl-D-aspartate; NQO-1: NAD (P)H: quinone oxidoreductase-1;OGD: oxygen-glucose deprivation; OGD/R: oxygen and glucose deprivation/reoxygenation; OVA: ovariectomy; PGC-1α: peroxisome proliferator-activated receptor-gamma coactivator 1α; PI3K: phosphatidylinositol 3-kinase; PPARγ, peroxisome proliferator-activated receptor γ; STAT3: signal transducer and activator of transcription 3; STZ: streptozotocin; TIRAP: Toll/interleukin-1 receptor domain-containing adapter protein; TLR4: Toll-like receptor 4; Trk: tyrosine kinase receptor.

## Data Availability

Not applicable.

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
