# Peer review of "Therapeutic Potential and Mechanisms of Novel Simple O-Substituted Isoflavones against Cerebral Ischemia Reperfusion"

_ijms, 2022, doi:10.3390/ijms231810394_

Round 1

Reviewer 1 Report

In the manuscript entitled “Therapeutic potential and mechanisms of novel simple O-substituted isoflavones against cerebral ischemia reperfusion”, Yang and colleagues describe the biosynthesis, metabolism, and neuroprotective mechanism of isoflavones on cerebral ischemia reperfusion.

The review is well written and informative, and lists all the relevant literature about this interesting topic.

However, some issues need to be addressed before publication.

In details:

1.      The authors, correctly, categorized stroke into ischemic and hemorrhagic (section 1, Introduction). They should also mention the subarachnoid hemorrhage as subtype of hemorrhagic stroke.

2.   The authors should discuss in the introduction (section 1) the (very limited) treatment options available against stroke.

3.      When describing studies on animals subjected to MCAO, the authors should specify which MCAO model are referring to (Transient? Permanent?)

4.      The authors, analysing the anti-stroke properties of genistein, described the effect of the molecule in neonatal mice (line 420). Since stroke can affect both mature and immature brains, the authors should discuss also perinatal stroke in the introduction section.

5.    The authors used the term “microflora” describing the metabolism of isoflavones occurring at intestinal level (section 3, The metabolism of isoflavones, page 10). The term is not correct and should be replaced with “microbiota” or “microorganisms” [for a definition of gut microorganisms see “Marchesi et al., Microbiome. 2015 3:31” and “Berg et al., Microbiome. 2020 8(1):103”].

6.      Figure 1 is not on the same level of the other figures. Does the circles overlapping mean an interaction between the pathophysiological mechanisms of cerebral ischemia reperfusion injury? Please explain and modify (or delete) the figure.

7.      A table listing the analysed isoflavones, the experimental models employed, the applied dosage and the length of application, the observed effects, and the proper references, could help the visualization of the isoflavones properties.

Minor:

1.      A reference to figure 1 is missing in the text.

2.      In figure 4, the words “glyceollin I, II and III” should lay on the same line.

Author Response

Dear,

I would like to express our thanks for your tremendous efforts in reviewing our paper. We have modified the manuscript accordingly, and the detailed corrections are listed below point by point:

Comment #1) The authors, correctly, categorized stroke into ischemic and hemorrhagic (section 1, Introduction). They should also mention the subarachnoid hemorrhage as subtype of hemorrhagic stroke.

Thank you very much for your suggestion. I have paid attention to the literature search while writing this manuscript. Only one of all the collected literature indicates that biochanin reduced neuronal apoptosis on subarachnoid hemorrhage. I have added the statement about subarachnoid hemorrhage in the Introduction.

Comment #2) The authors should discuss in the introduction (section 1) the (very limited) treatment options available against stroke.

Thank you very much for your suggestion. I have added the statement about drug therapy in the Introduction.

Comment #3) When describing studies on animals subjected to MCAO, the authors should specify which MCAO model are referring to (Transient? Permanent?)

Thank you very much for your suggestion. I have specified which MCAO model are referring to (Transient? Permanent?) in the main text.

Comment #4) The authors, analysing the anti-stroke properties of genistein, described the effect of the molecule in neonatal mice (line 420). Since stroke can affect both mature and immature brains, the authors should discuss also perinatal stroke in the introduction section.

Thank you very much for your suggestion. I have added the statement about perinatal stroke in the Introduction.

Comment #5) The authors used the term “microflora” describing the metabolism of isoflavones occurring at intestinal level (section 3, The metabolism of isoflavones, page 10). The term is not correct and should be replaced with “microbiota” or “microorganisms” [for a definition of gut microorganisms see “Marchesi et al., Microbiome. 2015 3:31” and “Berg et al., Microbiome. 2020 8(1):103”].

Thank you very much for your suggestion. I have corrected it.

Comment #6) Figure 1 is not on the same level of the other figures. Does the circles overlapping mean an interaction between the pathophysiological mechanisms of cerebral ischemia reperfusion injury? Please explain and modify (or delete) the figure.

Thank you very much for your suggestion. I have deleted it.

Comment #7) A table listing the analysed isoflavones, the experimental models employed, the applied dosage and the length of application, the observed effects, and the proper references, could help the visualization of the isoflavones properties.

Thank you very much for your suggestion. I have added one table in the main text.

Comment #8) A reference to figure 1 is missing in the text.

Thank you very much for your suggestion. I have deleted figure 1.

Comment #9) In figure 4, the words “glyceollin I, II and III” should lay on the same line.

Thank you very much for your suggestion. I have corrected it.

Sincerely Yours,

Chi-Rei Wu

Department of Chinese Pharmaceutical Sciences and Chinese Medicine Resources

China Medical University

[email protected]

Reviewer 2 Report

This is quite a good written paper.

The subject is very interesting and compatible with the journal scope. 

In a well-written introduction, an additional clinical paragraph is needed. The clinical problems are not part of the paper but a few sentences in the introduction should be added.

Figure 4 should be re-edited to fit on one page (with description). In its present form is very hard to read and understand.

Referenced at position 114. the doi: is not in the correct format - this is the hyperlink to the article.

Author Response

Dear,

I would like to express our thanks for your tremendous efforts in reviewing our paper. We have modified the manuscript accordingly, and the detailed corrections are listed below point by point:

Comment #1) This is quite a good written paper.

Thank you very much.

Comment #2) The subject is very interesting and compatible with the journal scope.

Thank you very much.

Comment #3) In a well-written introduction, an additional clinical paragraph is needed. The clinical problems are not part of the paper but a few sentences in the introduction should be added.

Thank you very much for your suggestion. I have added the statement about clinical management and drug therapy in the Introduction.

Comment #4) Figure 4 should be re-edited to fit on one page (with description). In its present form is very hard to read and understand.

Thank you very much for your suggestion. I have corrected figure 4 to fit on one page.

Comment #5) Referenced at position 114. the doi: is not in the correct format - this is the hyperlink to the article.

Thank you very much for your suggestion. I have corrected it.

Sincerely Yours,

Chi-Rei Wu

Department of Chinese Pharmaceutical Sciences and Chinese Medicine Resources

China Medical University

[email protected]

Reviewer 3 Report

I think authors have written an interesting narrative review about the role of simple O-substituted isoflavones in the context of cerebral ischemia. However, they could have performed a much more enriched systematic review or at least introduce quantitative/summarized analysis regarding up/downregulation of the different pathways. Moreover, the review lacks future perspectives and indications regarding the pitfalls in the field that should be addressed. Additionally, the text is too dense and needs serious grammar check, in my view.

Major revisions:

- The importance of the review is clarified by the authors in the abstract and seems fair, since there aren’t many reports like this.

- The review needs further sub-sections in order to be more readable and easier to follow. Instead of big chunks of text.

- In my view the abstract and the manuscript needs more clear conclusions. Meaning what are the pitfalls in the field, what are the future perspectives, what needs to be done! For example, are there any studies with non-primates or clinical trials? Successful, unsuccessful, underway, related?

- It would be also important for the authors to mention the word search performed, as well as the databases and dates used, similar to the systematic review analysis, to understand which manuscript were analyzed or considered for this narrative review.

- As mentioned a systematic review (or at least some table summary) would for sure enrich the manuscript! Authors could perform this kind of analysis, or at least do some kind of quantitative analysis, not just a narrative review. Authors should clearly quantify and indicate in summarizing tables the sensitivity, specificity and up/downregulation of the different pathways upon each type of isoflavone.

- The review needs important grammar revision. There a lot of errors.

- The figures are interesting, just missing the quantifying aspects.

- Abstract should be rewritten according to previous points.

Author Response

Dear,

I would like to express our thanks for your tremendous efforts in reviewing our paper. We have modified the manuscript accordingly, and the detailed corrections are listed below point by point:

Comment #1) The importance of the review is clarified by the authors in the abstract and seems fair, since there aren’t many reports like this.

Thank you very much.

Comment #2) The review needs further sub-sections in order to be more readable and easier to follow. Instead of big chunks of text.

Thank you very much for your suggestion. According to the reviewer's suggestion, the sections (4, 5 and 6) of genistein, daidzein and biochanin A are further divided into sub-sections due to their more studies and contents.

Comment #3) In my view the abstract and the manuscript needs more clear conclusions. Meaning what are the pitfalls in the field, what are the future perspectives, what needs to be done! For example, are there any studies with non-primates or clinical trials? Successful, unsuccessful, underway, related?

Thank you very much for your suggestion. I have rewritten the conclusion of Abstract section.

Comment #4) It would be also important for the authors to mention the word search performed, as well as the databases and dates used, similar to the systematic review analysis, to understand which manuscript were analyzed or considered for this narrative review.

Thank you very much for your suggestion. I have added the methodological statement in the first paragraph of each section of simple O-Substituted Isoflavones (4 - 8).

Comment #5) As mentioned a systematic review (or at least some table summary) would for sure enrich the manuscript! Authors could perform this kind of analysis, or at least do some kind of quantitative analysis, not just a narrative review. Authors should clearly quantify and indicate in summarizing tables the sensitivity, specificity and up/downregulation of the different pathways upon each type of isoflavone.

Thank you very much for your suggestion. I have added one table in the Conclusion section (9).

Comment #6) The review needs important grammar revision. There a lot of errors.

Thank you very much for your suggestion. Grammatical and writing style errors in the original version have been corrected by MDPI English editing services.

Comment #7) The figures are interesting, just missing the quantifying aspects.

Thank you very much for your suggestion. I have added one table in the Conclusion section (9).

Comment #8) Abstract should be rewritten according to previous points.

Thank you very much for your suggestion. I have rewritten the conclusion of Abstract section.

Sincerely Yours,

Chi-Rei Wu

Department of Chinese Pharmaceutical Sciences and Chinese Medicine Resources

China Medical University

[email protected]

Round 2

Reviewer 3 Report

Authors were able to improve the manuscript, following some of the suggestions. In my view the manuscript is now more clear for the readers and worthwhile publishing.